# Fibrodysplasia ossificans progressiva mutant ACVR1 signals by multiple modalities in the developing zebrafish

Robyn S Allen[1,2], Benjamin Tajer[1], Eileen M Shore[2], Mary C Mullins[1]*

[1]Department of Cell and Developmental Biology University of Pennsylvania Perelman School of Medicine, Philadelphia, United States; [2]Departments of Orthopaedic Surgery and Genetics University of Pennsylvania Perelman School of Medicine, Philadelphia, United States

**Abstract** Fibrodysplasia ossificans progressiva (FOP) is a rare human genetic disorder characterized by altered skeletal development and extraskeletal ossification. All cases of FOP are caused by activating mutations in the type I BMP/TGFβ cell surface receptor ACVR1, which over-activates signaling through phospho-Smad1/5 (pSmad1/5). To investigate the mechanism by which FOP-ACVR1 enhances pSmad1/5 activation, we used zebrafish embryonic dorsoventral (DV) patterning as an assay for BMP signaling. We determined that the FOP mutants ACVR1-R206H and -G328R do not require their ligand binding domain to over-activate BMP signaling in DV patterning. However, intact ACVR1-R206H has the ability to respond to both Bmp7 and Activin A ligands. Additionally, BMPR1, a type I BMP receptor normally required for BMP-mediated patterning of the embryo, is dispensable for both ligand-independent signaling pathway activation and ligand-responsive signaling hyperactivation by ACVR1-R206H. These results demonstrate that FOP-ACVR1 is not constrained by the same receptor/ligand partner requirements as WT-ACVR1.

**\*For correspondence:**
mullins@pennmedicine.upenn.edu

**Competing interests:** The authors declare that no competing interests exist.

## Introduction

Fibrodysplasia ossificans progressiva (FOP) is a rare human genetic disorder characterized by altered skeletal development and extraskeletal bone formation. Patients with FOP have pathognomonic malformation of the great toes and progressive spontaneous and injury-induced heterotopic ossification (HO) that eventually leads to loss of mobility (*Kaplan et al., 2009*; *Cohen et al., 1993*; *Connor and Evans, 1982*). Most cases of FOP are caused by a single amino acid substitution, R206H, in the type I BMP/TGFβ cell surface receptor ACVR1 (also known as ALK2), which over-activates signaling through phospho-Smad1/5 (pSmad1/5) (*Shore et al., 2006*; *Shen et al., 2009*; *van Dinther et al., 2010*; *Fukuda et al., 2009*). A small subset of patients with variant presentation of the classical FOP phenotype have distinct activating mutations in the *ACVR1* gene, including the substitution G328R (*Kaplan et al., 2009*; *Haupt et al., 2018*). While the phenotypic consequences of increased ACVR1 signaling have been well characterized in both patients and animal models (*Casal et al., 2019*; *Pignolo et al., 2011*; *Chakkalakal and Shore, 2019*), the mechanism by which ACVR1 mutations lead to over-active signaling is less well understood.

ACVR1 and its signaling partners belong to the transforming growth factor beta (TGFβ) superfamily. The activity of ACVR1 is critical to several developmental processes including embryonic patterning and skeletal formation (*Derynck and Akhurst, 2007*; *Little and Mullins, 2006*). In the presence of ligand, ACVR1 complexes with other BMP receptors to signal. In zebrafish, Acvr1l (also known as Alk8), the zebrafish paralog to human and mouse ACVR1 (*Yelick et al., 1998*), forms a tetrameric receptor complex with one other type I BMP receptor, Bmpr1a or Bmpr1b (also known as Alk3 and Alk6, respectively), and two type II BMP receptors (*Little and Mullins, 2006*; *Ehrlich et al., 2011*;

*Yadin et al., 2016*). Receptor complex assembly allows the type II BMP receptors to phosphorylate the type I receptors at serine/threonine residues within the GS domain (*Schmierer and Hill, 2007*; *Shi and Massagué, 2003*). Phosphorylation of the type I BMP receptors results in a conformational change, allowing them to bind ATP and phosphorylate Smad1/5 to initiate downstream transcription (*Feng and Derynck, 2005*; *Chaikuad et al., 2012*; *Liu et al., 1996*).

The zebrafish embryo is an excellent genetically tractable in vivo vertebrate model for investigating the signaling mechanism of the ACVR1-FOP receptor (*Shen et al., 2009*; *Mucha et al., 2018*). In the early zebrafish embryo, BMP acts as a morphogen to pattern the dorsoventral (DV) axis in a process that is conserved throughout the animal kingdom (*Zinski et al., 2018*). High levels of BMP signaling specify ventral cell fates and intermediate signaling specifies lateral fates, while absence of signaling allows dorsal cell fate specification. The DV pattern is generated through a quantifiable pSmad1/5 signaling gradient within the gastrulating embryo that peaks ventrally and decreases dorsally (*Little and Mullins, 2006*; *Zinski et al., 2017*; *Zinski et al., 2019*; *Tucker et al., 2008*; *Mintzer et al., 2001*). Perturbations to this BMP signaling gradient in the developing embryo result in distinct, dose-dependent patterning phenotypes (*Figure 1b*). Over-activation of the BMP signaling pathway by FOP-ACVR1 causes ventralization, an excess of ventral cell fate specification at the expense of dorsal fates (*Shen et al., 2009*; *Mucha et al., 2018*), while loss of endogenous *acvr1l* expression leads to an opposite dorsalization. Loss of *acvr1l* in the zebrafish can be rescued by human *ACVR1*, demonstrating their conserved activity (*Shen et al., 2009*).

Some aspects of ACVR1-R206H signaling have been investigated. Previous work by our lab demonstrated that expression of ACVR1-R206H in the zebrafish embryo over-activates BMP signaling in the absence of Bmp2 and Bmp7, the obligatory patterning ligands of the developing zebrafish (*Shen et al., 2009*; *Little and Mullins, 2009*; *Nguyen et al., 1998*; *Dick et al., 2000*; *Schmid et al., 2000*). Surprisingly, ACVR1-R206H shows acquired responsiveness to novel ligands in cell culture and mouse models; most notably Activin A, a TGFβ superfamily ligand that normally signals through ACVR1b (also called ALK4) and pSmad2/3 (*Hatsell et al., 2015*; *Lees-Shepard et al., 2018*; *Hino et al., 2015*). While ACVR1-R206H has been shown to require its normal type II BMP receptor partners, BMPR2 and ACVR2a (*Hino et al., 2015*; *Bagarova et al., 2013*), it is unknown whether it retains a requirement for its type I BMP receptor partner, BMPR1. ACVR1-R206H has been shown to signal in the absence of BMPR1a or BMPR1b individually (*Hino et al., 2015*). However, the ability of FOP-ACVR1 to function in the absence of both BMPR1a and BMPR1b, which largely function redundantly (*Yoon et al., 2005*; *Wine-Lee et al., 2004*), has not been tested.

In this study, we used BMP-pSmad1/5 dose-dependent DV patterning of the developing zebrafish to assay for signaling activity of ACVR1-R206H and ACVR1-G328R in vivo. We show that ligand-binding domain-deficient ACVR1-R206H and -G328R can over-activate pSmad1/5 signaling, demonstrating that these mutant receptors have enhanced signaling activity in the absence of ligand binding. However, intact ACVR1-R206H shows hyperactive pSmad1/5 signaling in response to Bmp7 and Activin A ligands. We further determined that neither the ligand-independent nor the ligand-responsive signaling modalities of ACVR1-R206H require the partner type I BMP receptors that are necessary for signaling by wild-type ACVR1. These results demonstrate that the ACVR1-R206H and G328R receptors have acquired a fundamentally altered signaling mechanism.

## Results

### ACVR1-R206H and -G328R over-activate BMP signaling in the absence of an intact ligand binding domain

Previous work by our lab showed that ACVR1-R206H can signal independently of Bmp2/7 heterodimers, the only functional DV patterning ligand in the zebrafish embryo (*Little and Mullins, 2009*), suggesting that the mutant receptor can signal independently of all BMP ligand (*Shen et al., 2009*). More recent studies have reported that ACVR1-R206H has an acquired response to the TGFβ family ligand, Activin A (*Hatsell et al., 2015*; *Lees-Shepard et al., 2018*; *Hino et al., 2017*). This ligand response has been implicated in inciting heterotopic ossification in patients with FOP (*Hino et al., 2017*; *Alessi Wolken et al., 2018*). Interestingly, however, ACVR1-R206H can signal without its ligand-binding domain in *Drosophila* and murine cell culture systems (*Haupt et al., 2018*; *Le and Wharton, 2012*; *Hildebrand et al., 2017*; *Haupt et al., 2014*).

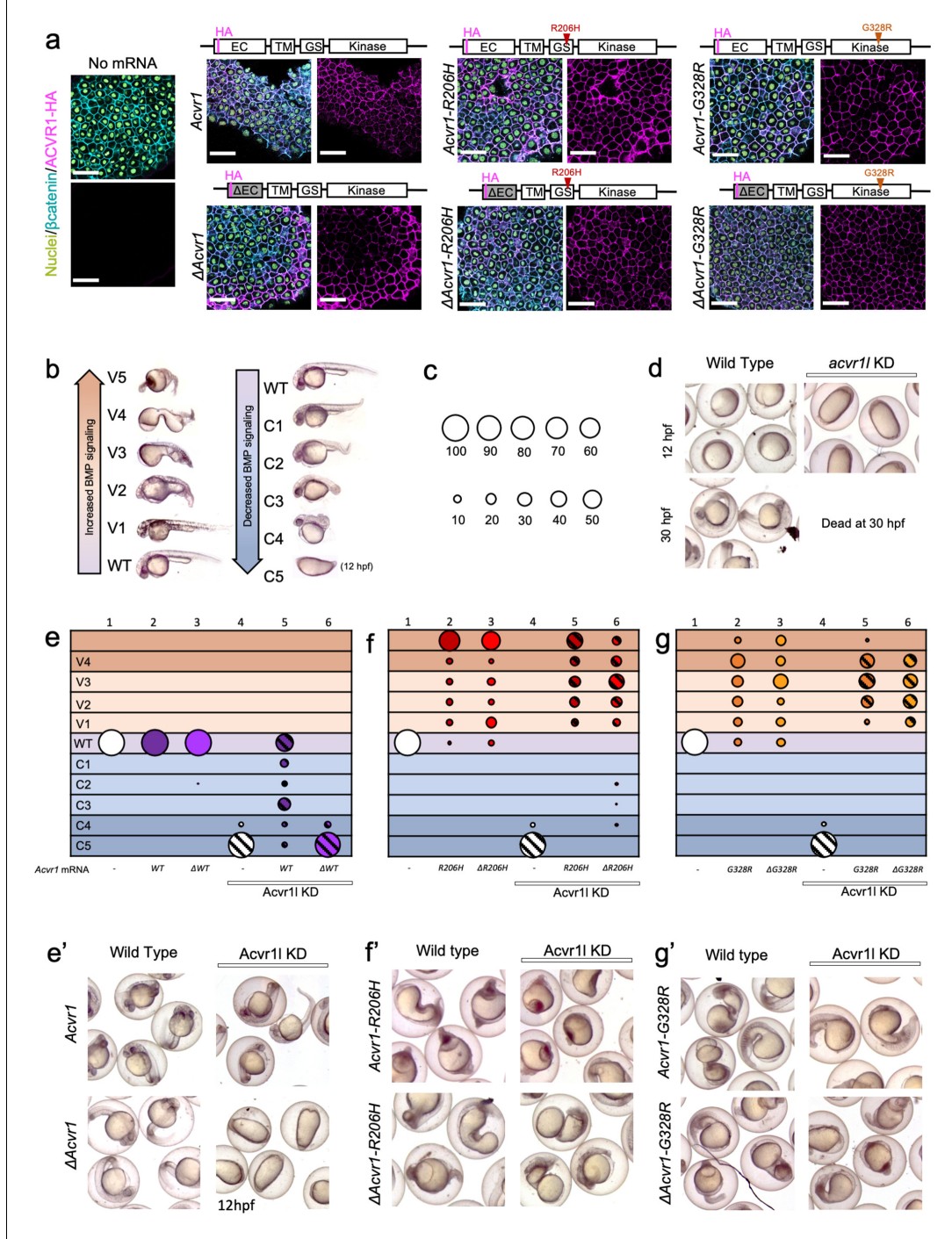

**Figure 1.** ACVR1-R206H and -G328R over-activate BMP signaling in the absence of an intact ligand-binding domain. (a) Embryos expressing ACVR1 HA-tagged constructs were immunostained for HA at the shield stage (early gastrula); ACVR1 domain schematics of the constructs are above each set of corresponding images. Nuclei (yellow), βcatenin (cyan), ACVR1-HA (magenta). Scale bars = 40 μm. ΔAcvr1 constructs lack residues 35–100, which contain the cysteine-rich ligand binding motifs. Uninjected (N = 6) and embryos injected with: *Acvr1* (N = 6), Δ*Acvr1* (N = 5), *Acvr1-R206H* (N = 9), Δ*Acvr1-R206H* (N = 4), *Acvr1-G328R* (N = 5), and Δ*Acvr1-G328R* (N = 5). (b) BMP signaling dose-dependent 12–30 hpf zebrafish embryo phenotypes: severe dorsalization (C5-C4, dark blue), mild dorsalization (C3-C1, light blue), wild-type development (WT, violet), mild ventralization (V1-V3, light orange), and severe ventralization (V4-V5, dark orange). (c) Bubble plot circle sizes correspond to the percent of total embryos within a condition that are in a particular phenotypic category. Examples are shown for: 100%, 90%, 80%, 70%, 60%, 50%, 40%, 30%, 20% and 10%. (d) Representative phenotypes of wild-type embryos (evaluated at 12 and 30 hpf) and embryos injected with *acvr1l* MO (evaluated at 12 hpf; none survive to 30 hpf). (e–g) DV phenotypes of 12–30 hpf embryos injected with *Acvr1* or Δ*Acvr1* mRNA alone or together with *acvr1l* MO. Data are from four pooled experiments.

*Figure 1 continued on next page*

Figure 1 continued

(e) *Acvr1* or *ΔAcvr1* injected embryos. Columns: 1, N = 240; 2, N = 121; 3, N = 140; 4, N = 153; 5, N = 96; 6, N = 118. (e') Representative 12 or 30 hpf phenotypes. (f) *Acvr1-R206H* or *ΔAcvr1-R206H* injected embryos. Columns: 1, N = 240; 2, N = 70; 3, N = 84; 4, N = 153; 5, N = 73; 6, N = 91. (f') Representative 30 hpf phenotypes. (g) *Acvr1-G328R* or *ΔAcvr1-G328R* injected embryos. Columns: 1, N = 240; 2, N = 88; 3, N = 79; 4, N = 153; 5, N = 78; 6, N = 71. (g') Representative 30 hpf phenotypes.

The online version of this article includes the following source data and figure supplement(s) for figure 1:

**Source data 1.** Injected embryo phenotype raw numbers for *Figure 1e, f and g*.
**Figure supplement 1.** ACVR1-R206H and -G328R over-activate BMP signaling causing ventralization in the absence of an intact ligand-binding domain.
**Figure supplement 2.** Ventralizing amounts of Acvr1-R206H and Acvr1-G328R are comparable to rescuing amounts of Acvr1.
**Figure supplement 2—source data 1.** Injected embryo phenotype raw numbers for *Figure 1—figure supplement 2a*.
**Figure supplement 2—source data 2.** Beta-catenin surface area and sum HA-tag fluorescence for *Figure 1—figure supplement 2b*.

To investigate the FOP-ACVR1 signaling mechanism, we used a zebrafish DV patterning assay. For this assay, we microinjected one-cell stage wild-type or BMP component mutant zebrafish embryos to introduce mRNAs and/or anti-sense knockdown morpholinos that replicate null mutant phenotypes. The mRNAs and morpholinos were never mixed together, but instead each mRNA and morpholino was injected using separate needles, so that controls could be done to show the efficacy of each component on its own. Within one experiment, each mRNA or morpholino was always injected through the same calibrated needle on its own injection apparatus to ensure consistent amounts were injected among all the controls and experimental conditions.

To test if the FOP-ACVR1 receptor requires ligand binding to signal in a vertebrate animal model, we first compared the signaling function of the mouse wild-type ACVR1 to mouse ACVR1 lacking 65 residues containing the cysteine-rich ligand-binding motifs of the extracellular domain (ΔACVR1) (*Haupt et al., 2014*). We injected one-cell stage embryos with *Acvr1* or *ΔAcvr1* mRNA and tested if the exogenously expressed ACVR1 protein correctly localized to the cell membrane by immunostaining for the HA-epitope tag present on these receptors. We found that all the ACVR1 protein variants, regardless of their signaling activity, were expressed and localized to the cell membrane within the developing embryo (*Figure 1a*).

To determine if the ligand-binding domain mutant ΔACVR1 receptor was sufficient to pattern the zebrafish embryo, we knocked down endogenous zebrafish Acvr1l with morpholinos (MOs), which replicate the phenotype of *acvr1l* null maternal-zygotic mutant embryos (*Mintzer et al., 2001*). Although we have previously generated maternal-zygotic *acvr1l* mutant embryos that lack both maternally and zygotically expressed *acvr1l*, are deficient in all BMP signaling, and exhibit a C5 severely dorsalized phenotype (*Mintzer et al., 2001*), generating these maternal-zygotic *acvr1l* mutant embryos required producing *acvr1l* homozygous mutant adults, an arduous accomplishment due to other requirements for Acvr1l later in development (*Mintzer et al., 2001*). Therefore, we used MO knockdown in the current study to deplete Acvr1l on a large scale and in combination with other mutant genotypes. As previously demonstrated and re-confirmed here, knockdown of Acvr1l generates the identical, severely dorsalized C5 phenotype as the maternal-zygotic *acvr1l* null mutant, consistent with the loss of all BMP pathway activity (*Figure 1d* and column four in *Figure 1e–f*; *Figure 1—figure supplement 1*; *Mintzer et al., 2001*; *Little and Mullins, 2009*; *Bauer et al., 2001*). At 12 to 14 hr post fertilization (hpf), the C5 phenotype is characterized by an elongated body axis and by expansion of the normally dorsally-located somites around the circumference of the embryo. By 30 hpf, these embryos lyse due to presumptive pressure from the radialized somites. Importantly, injection of wild-type mouse *Acvr1* mRNA can fully rescue this phenotype (*Figure 1e,e'*, compare to d), demonstrating the specificity of the knockdown for Acvr1l. All these control experiments adhere to the recently published guidelines by leaders in the zebrafish field for the use of MOs and their validity to substitute for a mutant allele (*Stainier et al., 2017*).

We next expressed *Acvr1* or *ΔAcvr1* mRNAs in *acvr1l*-knockdown (*acvr1l*-KD) embryos to test their function. We evaluated rescue of BMP signaling activity in these embryos by assaying for DV patterning phenotypes (*Figure 1b*) and quantifying the proportion of embryos in each phenotypic category (*Figure 1c*). Neither *Acvr1* nor *ΔAcvr1* mRNA perturbed normal development in the presence of endogenous Acvr1l (*Figure 1e* columns 2 and 3, *Figure 1e'*; *Figure 1—figure supplement 1*). While ACVR1 rescued loss of endogenous Acvr1l, primarily to wild-type or mildly dorsalized phenotypes (*Figure 1e* column 5, *Figure 1e'*; *Figure 1—figure supplement 1*), *ΔAcvr1* did not

(*Figure 1e* column 6, and *Figure 1e'*; *Figure 1—figure supplement 1*). By contrast, both *Acvr1-R206H* and *ΔAcvr1-R206H* ventralized WT and *acvr1l*-KD zebrafish embryos (*Figure 1f* columns 2, 3, 5 and 6, *Figure 1f'*; *Figure 1—figure supplement 1*). Likewise, both *Acvr1-G328R* and *ΔAcvr1-G328R* ventralized WT and *acvr1l*-KD embryos (*Figure 1g* columns 2, 3, 5 and 6, *Figure 1g'*; *Figure 1—figure supplement 1*). These results in an in vivo vertebrate model support that ACVR1-R206H and ACVR1-G328R, unlike wild type ACVR1, exhibit ligand-independent signaling activity.

To test if our results could be influenced by differences in expression levels of injected mRNA, we determined the relative amounts of receptor in our injected embryos, using immunofluorescence to measure the cell surface expression of our Acvr1-HA-tagged mutant receptors in confocal sections of mid-gastrula embryos. Mid-gastrula embryos were collected for immunofluorescence and the remaining embryos were allowed to develop to 30 hpf for phenotyping. As described above, Acvr1, but not ΔAcvr1 could rescue Acvr1l-KD embryos, and all FOP mutant receptors ventralized embryos regardless of the presence of a ligand-binding domain (*Figure 1—figure supplement 2*). The amount of WT-Acvr1 that partially to fully rescued Acvr1l KD was determined per µm$^2$ of β-catenin fluorescence surface area (*Figure 1—figure supplement 2*). This receptor amount should reflect the amount of receptor expression that is required to pattern the zebrafish embryo. The amounts of Acvr1-R206H and Acvr1-G328R fluorescence that ventralized the embryo were similar to or significantly lower than WT-Acvr1, respectively. The levels of ΔAcvr1-G328R and ΔAcvr1-R206H fluorescence were also similar to or significantly lower than ΔAcvr1, respectively. These results show that the amount of Acvr1-R206H or -G328R required to ventralize the zebrafish embryo is similar to or lower than the amount of Acvr1 required to pattern the wild-type zebrafish, regardless of the presence of a ligand-binding domain. Altogether, the results indicate that the ventralization is not due to overexpression of the FOP mutant receptors.

## ACVR1-R206H and -G328R over-activate pSmad1/5 signaling with or without the presence of a ligand-binding domain

During early embryonic development, a nuclear gradient of pSmad1/5 activity forms across the DV axis of the zebrafish embryo in response to BMP signaling (*Zinski et al., 2017*; *Zinski et al., 2019*; *Tucker et al., 2008*; *Figure 2a*). This gradient persists throughout gastrulation and specifies DV axial fates. To test the ability of the ligand-binding domain mutant ΔAcvr1 to signal through pSmad1/5, we knocked down endogenous Acvr1l, injected embryos with mouse *Acvr1* or *ΔAcvr1* mRNAs, and then immunostained early-gastrula embryos (shield-65% epiboly stage) for pSmad1/5.

While wild-type embryos formed a gradient of pSmad1/5 expression that peaks ventrally and decreases dorsally (*Figure 2a*; *Mucha et al., 2018*), Acvr1l-KD embryos lacked detectable pSmad1/5 signal (*Figure 2a'*). The mean nuclear pSmad1/5 fluorescence of Acvr1l-KD embryos was significantly decreased compared to WT embryos (*Figure 2h*). Injected mouse *Acvr1* mRNA did not alter the pSmad1/5 gradient in WT embryos and rescued pSmad1/5 in Acvr1l-KD embryos (*Figure 2b,b'*). There was no significant difference in mean nuclear pSmad1/5 intensity between uninjected wild-type, *Acvr1*-injected wild-type, and Acvr1l-KD embryos injected and rescued with *Acvr1* (*Figure 2h*). *ΔAcvr1* also did not perturb normal gradient formation in WT embryos, but could not rescue loss of pSmad1/5 in Acvr1l-KD embryos (*Figure 2c,c'*). There was no significant difference in mean pSmad1/5 fluorescence between Acvr1l-KD embryos and Acvr1l-KD embryos injected with *ΔAcvr1*, showing that WT-Acvr1 cannot signal through pSmad1/5 without a ligand-binding domain (*Figure 2h*). In contrast, both *Acvr1-R206H* and *ΔAcvr1-R206H* restored pSmad1/5 signaling in Acvr1l-KD embryos and greatly expanded the signaling gradient dorsally (*Figure 2d,d',e,e'*). Likewise, Acvr1-G328R and ΔAcvr1-G328R rescued and expanded the pSmad1/5 gradient even when endogenous Acvr1l was absent (*Figure 2f,f',g,g'*). Both ACVR1-R206H and ACVR1-G328R significantly increased the mean pSmad1/5 signaling intensity in Acvr1l-KD embryos regardless of the presence of a ligand-binding domain (*Figure 2h*). These results confirm that ACVR1-R206H and ACVR1-G328R have acquired the capacity to over-activate Smad1/5 phosphorylation even in the absence of an intact ligand-binding domain.

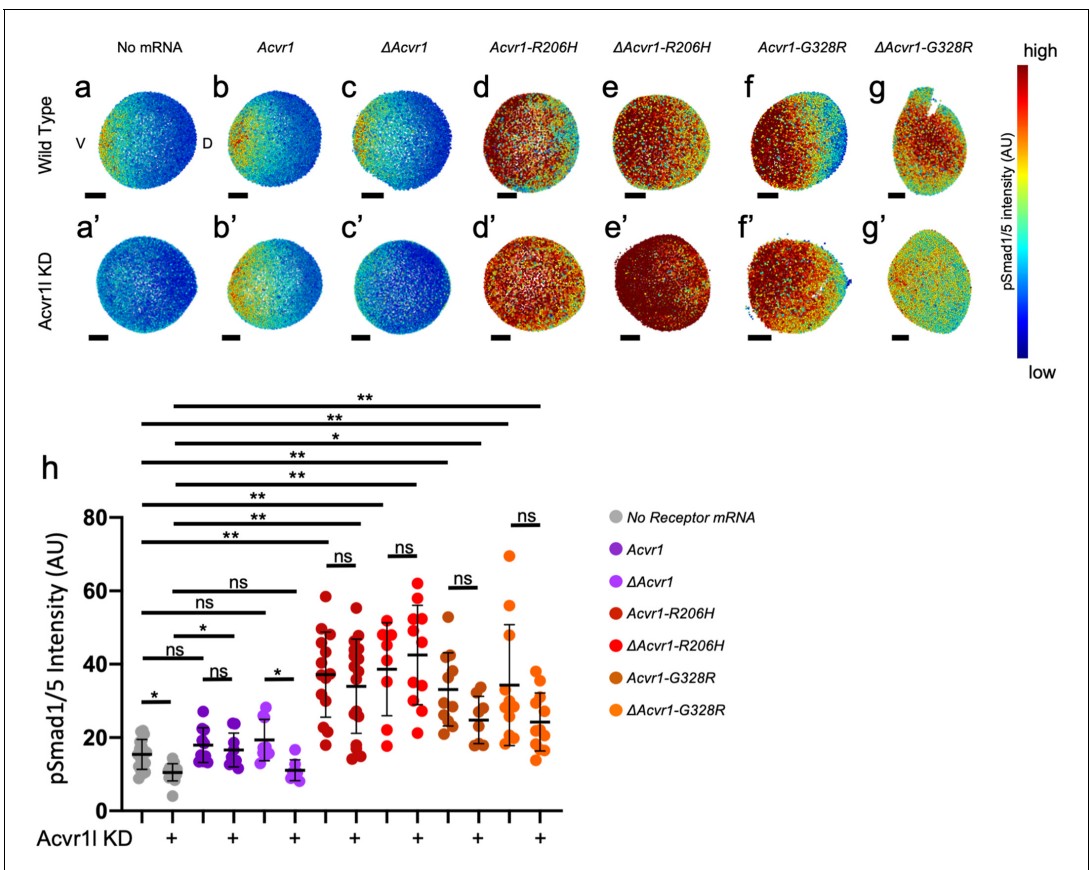

**Figure 2.** ACVR1-R206H and -G328R over-activate pSmad1/5 signaling with or without the presence of a ligand-binding domain. (a,a'–g,g') Animal pole view of relative pSmad1/5 intensities (using arbitrary units (AU)) within each nucleus of the embryo in representative WT early-gastrula embryos (shield-65% epiboly stage) with endogenous Acvr1l present (a–g) or knocked down (a'–g'). Injected mRNAs are noted above for b-g and b'-g'. (a–c, a'–c') Embryos are oriented with ventral (V) side to the left and the dorsal (D) side to the right. (d–g, d'–g') Dorsal side of the embryo could not be identified due to loss of the shield structure with severe ventralization. Scale bars = 100 μm. (a, a') Wild-type and Acvr1l-KD embryos, respectively (N = 18 and 15). (b, b') *Acvr1* injected embryos (N = 10 and 9). (c, c') *ΔAcvr1* injected embryos (N = 9 and 7). (d, d') *Acvr1-R206H* injected embryos (N = 14 and 17). (e, e') *ΔAcvr1-R206H* injected embryos (N = 8 and 11). (f, f') *Acvr1-G328R* injected embryos (N = 12 and 9). (g, g') *ΔAcvr1-G328R* injected embryos (N = 12 and 12). (h) Mean nuclear pSmad1/5 fluorescence of injected embryos. Each dot represents the mean nuclear fluorescence for an individual embryo. Mean and standard deviation of each condition are shown by bars. * indicates p<0.05, ** indicates p<0.001, ns indicates no significance.

The online version of this article includes the following source data for figure 2:

**Source data 1.** Average nuclear pSmad1/5 fluorescence for *Figure 1h*.
**Source data 2.** Raw nuclear pSmad1/5 fluorescence for *Figure 1h*.

## ACVR1-R206H and -G328R, but not ΔAcvr1-R206H, are responsive to Bmp7 ligand

We next tested if FOP-ACVR1 with an intact ligand-binding domain retained the ability to respond to ligand. We used Bmp7 in this experiment since previous studies showed that ACVR1 binds and signals in response to Bmp7 (*Yadin et al., 2016*; *Little and Mullins, 2009*; *Heinecke et al., 2009*; *Allendorph et al., 2007*). We injected human WT- or FOP-*ACVR1* mRNAs into one-cell stage *bmp7sb1aub* null mutant zebrafish embryos that also had Acvr1l KD. We then determined whether Bmp7 ligand expression enhanced signaling by these ACVR1 receptors.

Homozygous *bmp7-/-* embryos exhibit a severely dorsalized C5 phenotype (*Figure 3a,b* column 1; *Figure 3—figure supplement 1*; *Nguyen et al., 1998*; *Dick et al., 2000*; *Schmid et al., 2000*). This dorsalization can be rescued by injected *bmp7* mRNA (*Figure 3a,b* column 2; *Figure 3—figure supplement 1*). However, *bmp7* mRNA expression does not rescue *bmp7-/-* embryos that are also deficient for Acvr1l, as expected (*Figure 3a,b* column 3; *Figure 3—figure supplement 1*).

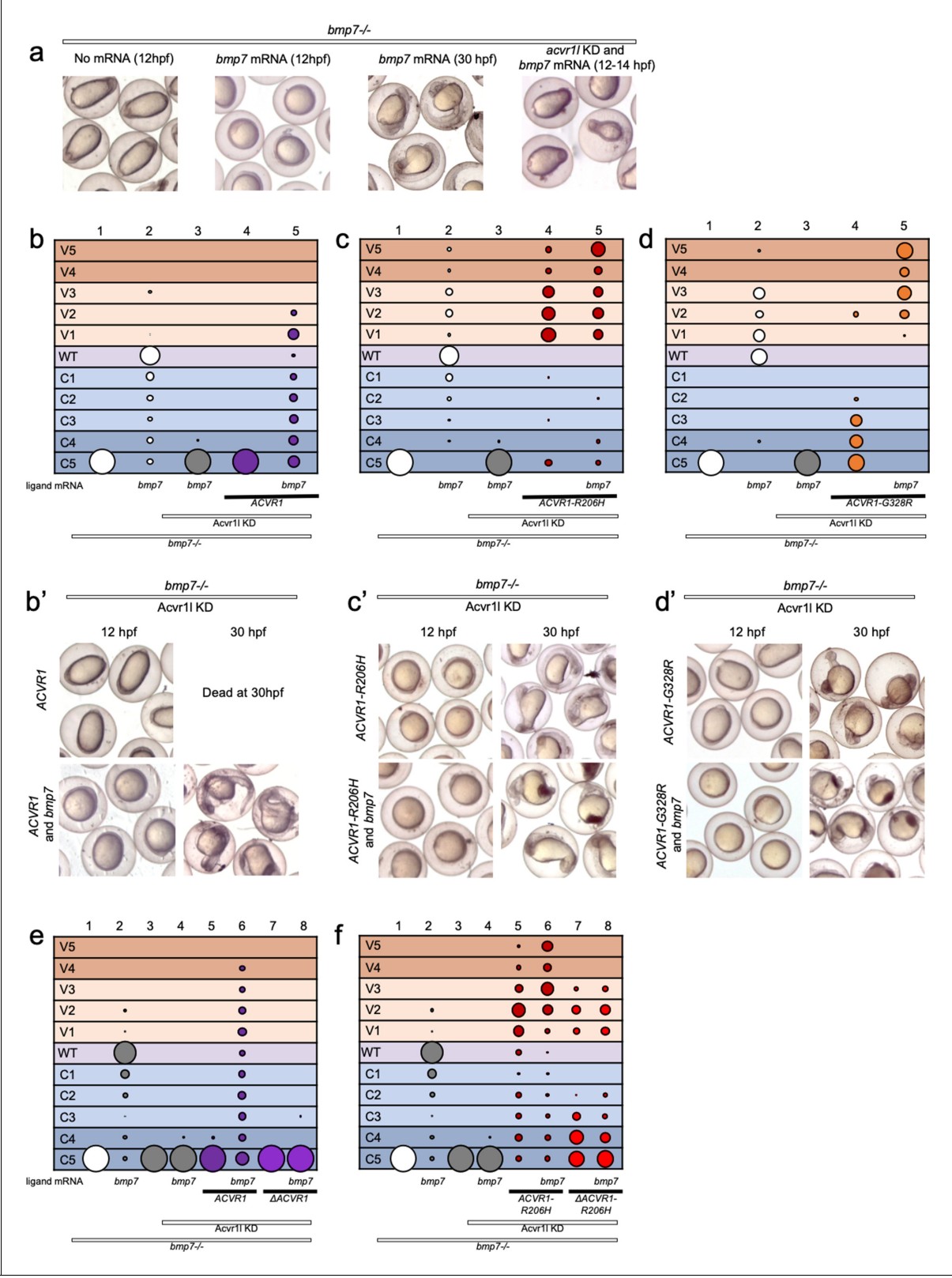

**Figure 3.** ACVR1-R206H and -G328R but not ΔAcvr1-R206H are responsive to Bmp7 ligand. (a) Representative dorsal-ventral (DV) phenotypes of *bmp7-/-* embryos not injected (12 hpf), injected with *bmp7* mRNA (12 and 30 hpf), or *bmp7* mRNA and *acvr1l* KD (12–14 hpf). (b–d) 12–30 hpf DV phenotypes of *bmp7-/-* embryos with *acvr1l* KD that were injected with human *ACVR1* mRNA alone or combined with *bmp7* mRNA. (b) WT *ACVR1* injected embryos. Three pooled experiments. Columns: 1, N = 240; 2, N = 150; 3, N = 143; 4, N = 101; 5, N = 120. (b') Representative 12 and 30 hpf

*Figure 3 continued on next page*

Figure 3 continued

phenotypes. (c) *ACVR1-R206H* injected embryos. Two pooled experiments. Columns: 1, N = 177; 2, N = 78; 3 N = 132; 4, N = 100; 5, N = 86. (c') Representative 12 and 30 hpf phenotypes. (d) *ACVR1-G328R* injected embryos. Two pooled experiments. Columns: 1, N = 52; 2, N = 46; 3, N = 55; 4, N = 56; 5, N = 82. (d') Representative 12 and 30 hpf phenotypes. (e–f) 12–30 hpf DV phenotypes of *bmp7-/-* embryos with *acvr1l* KD that were injected with a mouse *Acvr1* or *ΔAcvr1* mRNA alone or in combination with *bmp7* mRNA. Four pooled experiments. (e) WT *Acvr1* or *ΔAcvr1* mRNA Columns: 1, N = 540; 2, N = 172; 3, N = 143; 4, N = 152; 5, N = 231; 6, N = 163; 7, N = 104; 8, N = 97; (f) *Acvr1-R206H* or *ΔAcvr1-R206H* mRNA Columns: 1, N = 540; 2, N = 172; 3, N = 143; 4, N = 152; 5, N = 176; 6, N = 217; 7, N = 119; 8, N = 145.

The online version of this article includes the following source data and figure supplement(s) for figure 3:

**Source data 1.** Injected embryo phenotype raw numbers for *Figure 3b, c, d, e and f*.
**Figure supplement 1.** ACVR1-R206H and ACVR1-G328R are responsive to BMP7 ligand.
**Figure supplement 2.** Unlike Acvr1-R206H, neither ΔAcvr1-R206H nor ΔAcvr1 responds to Bmp7 ligand.

Conversely, in the absence of endogenous *bmp7*, WT ACVR1 expression cannot rescue *bmp7-/-* embryos (*Figure 3b* column 4, b'; *Figure 3—figure supplement 1*), confirming that both of these components are required for BMP signaling to pattern the developing embryo. Co-injection of human *ACVR1* with *bmp7* mRNA rescued embryos to less dorsalized or wild-type phenotypes (*Figure 3b* column 5, b'; Fig *Figure 3—figure supplement 1*). Similarly, mouse *Acvr1* rescued DV patterning with *bmp7* mRNA, but could not pattern embryos in the absence of *bmp7* (*Figure 3—figure supplement 1*, columns 5 and 6).

We next tested if ACVR1-R206H or ACVR1-G328R signaling can be enhanced by Bmp7 ligand. *ACVR1-R206H* rescued *bmp7-/-*, *acvr1l*-KD fish primarily to mildly ventralized phenotypes (*Figure 3c* column 4, c'; *Figure 3—figure supplement 1*). This rescue was enhanced to severe ventralization phenotypes by the addition of *bmp7* ligand mRNA (*Figure 3c* column 5, c', *Figure 3—figure supplement 1*). Similarly, *ACVR1-G328R* rescued *bmp7-/-* embryos to less severe dorsalized phenotypes (*Figure 3d* column 4, d', *Figure 3—figure supplement 1*) and induced further ventralization in response to *bmp7* ligand (*Figure 3d* column 5, d'; *Figure 3—figure supplement 1*).

We then tested if our intact and ligand-binding mutant mouse Acvr1 could respond to Bmp7. Like human ACVR1, mouse Acvr1 only rescued *bmp7-/-* mutant fish in the presence of injected *bmp7* mRNA (*Figure 3e*, columns 5 and 6; *Figure 3—figure supplement 2*). *ΔAcvr1*, however, could not rescue *bmp7-/-* embryos regardless of the presence of Bmp7 (*Figure 3e*, columns 7 and 8; *Figure 3—figure supplement 2*), demonstrating that WT Acvr1 requires a ligand-binding domain to signal. *Acvr1-R206H* rescued *bmp7-/-*, *acvr1l*-KD embryos primarily to mildly ventralized phenotypes (*Figure 3f*, column 5; *Figure 3—figure supplement 2*) and additional *bmp7* ligand mRNA enhanced the rescue to severely ventralized phenotypes (*Figure 3f*, column 6; *Figure 3—figure supplement 2*). Importantly, while *ΔAcvr1-R206H* rescued *bmp7-/-* embryos to ventralized phenotypes (*Figure 3e*, column 7; *Figure 3—figure supplement 2*), the addition of *bmp7* mRNA did not further enhance ventralization (*Figure 3e*, column 8; *Figure 3—figure supplement 2*), supporting that loss of the ligand-binding domain prevents ligand response. These data show that ACVR1-R206H and ACVR1-G328R are responsive to BMP ligand, and together with the data in *Figures 1* and *2*, demonstrate that these FOP-ACVR1 mutant receptors have both ligand-independent and ligand-responsive activity.

## ACVR1-R206H signaling in response to activin A ligand depends on its ligand binding domain

Studies have shown that ACVR1-R206H has acquired the ability to respond to Activin A ligand, in addition to its normal BMP ligands (*Hatsell et al., 2015*; *Lees-Shepard et al., 2018*; *Hino et al., 2015*). Previous studies reported that ΔAcvr1-R206H does not respond to Activin A or BMP ligand in cell culture (*Hildebrand et al., 2017*). To test if ACVR1-R206H requires a ligand-binding domain to activate pSmad1/5 in response to Activin A or BMP7 ligand in vivo in the zebrafish vertebrate model, we injected Acvr1l-KD, *bmp7-/-* embryos with mouse *Acvr1* or *ΔAcvr1* mRNA with or without *Activin A* or *bmp7* mRNA and assayed pSmad1/5 activation in six hpf embryos and DV phenotypes in 30 hpf embryos.

Expression of Activin A ligand in the early embryo enhances Nodal signaling through pSmad2 (*Figure 4p*) and dorsal organizer mesoderm formation (*Gritsman et al., 1999*; *Thisse et al., 2000*; *Green et al., 1992*; *Green et al., 1994*), precluding our ability to assay DV patterning defects. In

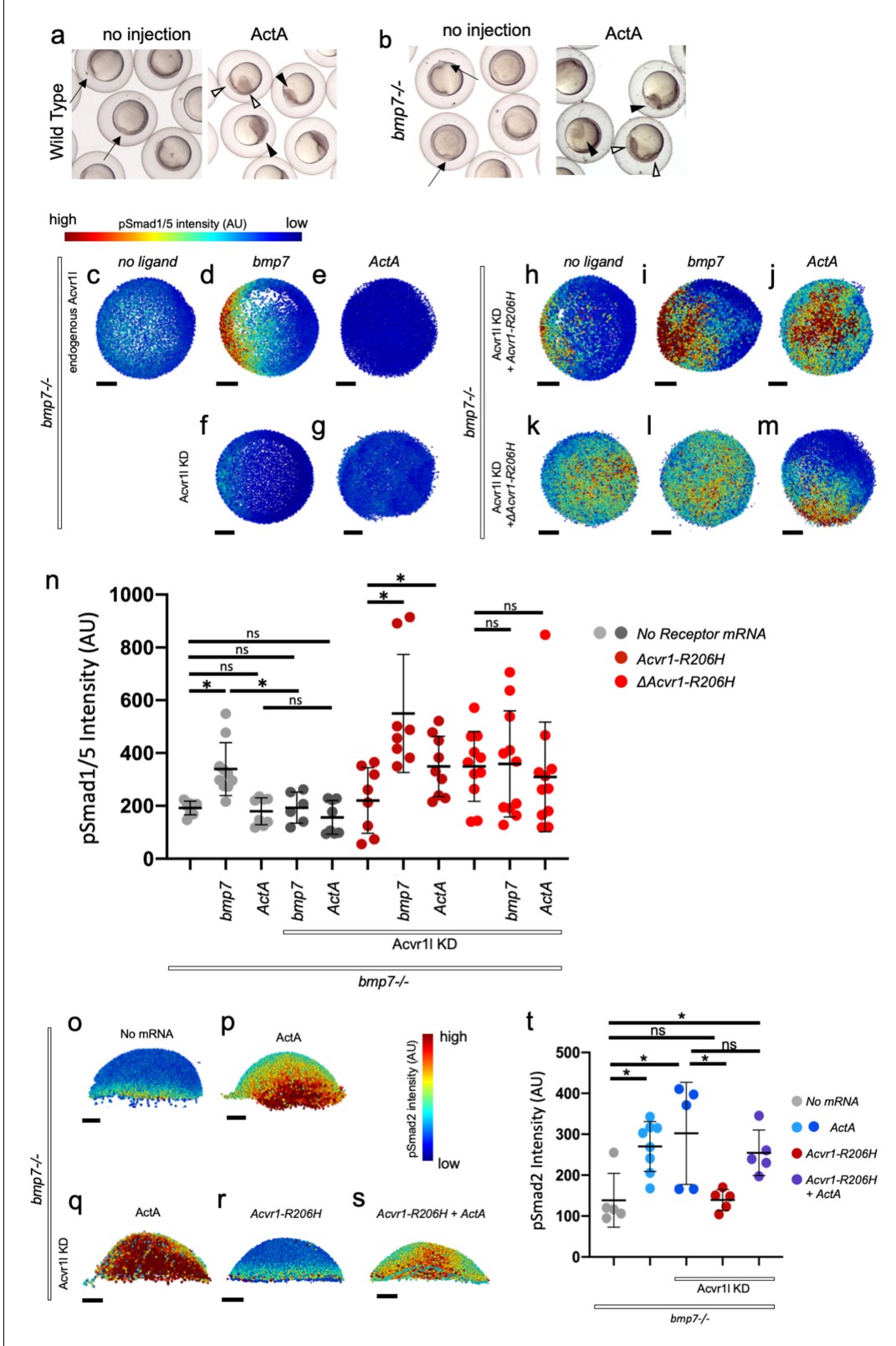

**Figure 4.** ACVR1-R206H requires a ligand-binding domain to respond to BMP7 and Activin A ligand through pSmad1/5. (a, b) Representative 6 hpf (shield stage) WT (a) and *bmp7-/-* (b) embryos uninjected or injected with *Activin A* mRNA. WT and *bmp7-/-* develop normal dorsal organizers (arrows). *Activin A* injection causes embryos to have expanded (filled arrowheads) or duplicated (empty arrowheads) dorsal organizers. (c–m) Animal pole view of pSmad1/5 intensities (using arbitrary units (AU)) within each nucleus of the embryo in representative *bmp7-/-* early gastrula (shield stage) embryos with

*Figure 4 continued on next page*

*Figure 4 continued*

endogenous Acvr1l present (**c–e**) or knocked down (**f–m**). Injected receptor mRNA is noted to the left and ligand mRNA is noted above. Embryos are oriented with the dorsal shield to the right. Scale bars = 100 µm. (**c–e**) *bmp7-/-* uninjected embryos (N = 8), or injected with *bmp7* (N = 10), or *Activin A* (N = 8) mRNA. (**f and g**) *bmp7-/-, acvr1l* KD embryos injected with *bmp7* (N = 6) or *Activin A* (N = 8) mRNA. (**h–j**) *Acvr1-R206H* injected *bmp7-/-, acvr1l* KD embryos with no injected ligand (N = 8), *bmp7* (N = 8), or *Activin A* (N = 9) mRNA. (**k–m**) *ΔAcvr1-R206H* injected *bmp7-/-, acvr1l* KD embryos (N = 11), with injected *bmp7* (N = 11), or *Activin A* (N = 11) mRNA. (**n**) Mean nuclear pSmad1/5 fluorescence of injected embryos. Each dot represents the mean fluorescence of an individual embryo. Mean and standard deviation of each condition is shown. * indicates p<0.05, ns indicates no significance. (**o–s**) Lateral views showing relative pSmad2 intensities (using arbitrary units (AU)) within each nucleus of the embryo in representative *bmp7-/-* early gastrula embryos (shield stage) with endogenous *acvr1l* present (**i and j**) or knocked down (**k–m**). Injected mRNA is noted above each image. Embryos are oriented with the presumptive dorsal side facing forward. Scale bars = 100 µm. (**o**) *bmp7-/-* embryo (N = 5) (**p**) *Activin A* injected embryo (N = 8) (**q**) *Activin A* injected embryo with Acvr1l knockdown (N = 5) (**r**) *mAcvr1-R206H* injected embryo with Acvr1l knockdown (N = 5) (**s**) *mAcvr1-R206H* and *Activin A* injected embryo with Acvr1l knockdown (N = 5) (**t**) Mean nuclear pSmad2 fluorescence of injected embryos. Each dot represents the mean fluorescence of an individual embryo. Mean and standard deviation of each condition is shown by bars. * indicates p<0.05, ns indicates no significance.

The online version of this article includes the following source data for figure 4:

**Source data 1.** Average nuclear pSmad1/5 fluorescence for *Figure 4n*.
**Source data 2.** Raw nuclear pSmad1/5 fluorescence for *Figure 4n*.
**Source data 3.** Average nuclear pSmad2 fluorescence for *Figure 4t*.
**Source data 4.** Raw nuclear pSmad2 fluorescence for *Figure 4t*.

wildtype and *bmp7-/-* embryos, the dorsal organizer or shield forms on the presumptive dorsal side of the early gastrula-stage embryo (*Figure 4a,b*). Overexpression of *Activin A* mRNA in the embryo results in duplication or expansion of the dorsal organizer regardless of the presence of *bmp7* (*Figure 4a,b*). This expansion of dorsal structures causes development to halt, leading to subsequent death or severe perturbation of patterning.

Therefore, to assess the ability of Acvr1-R206H to respond to Activin A with or without a ligand-binding domain, we used immunostaining to quantify pSmad1/5 activity in Acvr1l-KD, *bmp7-/-* embryos injected with *Acvr1* or *ΔAcvr1* with or without *bmp7* or *Activin A*. For these experiments, a lower amount of *Acvr1-R206H* and *ΔAcvr1-R206H* mRNA was injected than in the DV phenotyping experiments to decrease the initial pSmad1/5 intensity prior to the addition of ligand and permit detection of Activin A ligand-induced effects. Embryos were collected at shield stage before Activin A expressing embryos halt their development.

A Bmp2/7 heterodimer is the obligatory patterning ligand in the gastrula zebrafish embryo (*Little and Mullins, 2009*), and as a result, *bmp7-/-* early gastrula embryos lack detectable pSmad1/5 signaling (*Figure 4c*). Injection of *bmp7* mRNA rescues pSmad1/5 signaling to wild-type levels that peak ventrally and decrease dorsally (*Figure 4d*). Addition of *bmp7* mRNA did not rescue signaling in the absence of endogenous Acvr1l (*Figure 4f*). Addition of Activin A did not rescue pSmad1/5 signaling regardless of the presence of Acvr1l (*Figure 4e,g*). To quantitatively compare the pSmad1/5 signaling levels of the embryos in these conditions, we determined the mean pSmad1/5 intensities for each condition (*Figure 4n*). We found that Bmp7 significantly increased the mean pSmad1/5 signaling intensity compared to *bmp7-/-* embryos (*Figure 4n*). Expression of Bmp7 or Activin A in Acvr1l-KD embryos did not significantly alter the mean pSmad1/5 signaling level (*Figure 4n*), showing that Bmp7 requires Acvr1 to induce pSmad1/5 signaling and Activin A normally has no significant effect on pSmad1/5 signaling.

The low level of injected *Acvr1-R206H* mRNA activated low levels of pSmad1/5 signaling in *bmp7-/-* embryos (*Figure 4h*). Importantly, co-injection of *Acvr1-R206H* with *Activin A* or *bmp7* mRNA significantly increased the mean pSmad1/5 signaling intensity (*Figure 4i,j,n*). Our data suggest that ACVR1-R206H had a more intense response to Bmp7 than Activin A; however, it is important to note that the amount of translated ligand in an embryo was not determined in these experiments. In addition, the dorsal organizer expansion in response to Activin A in our experiments may mask pSmad1/5 intensity enhancement by ACVR1-R206H in response to Activin A. These data demonstrate that Acvr1-R206H activates pSmad1/5 signaling in response to both Activin A and Bmp7.

Conversely, while *ΔAcvr1-R206H* induced pSmad1/5 in *bmp7-/-* embryos (*Figure 4k,n*), neither addition of Activin A nor Bmp7 significantly increased the mean pSmad1/5 fluorescence in *ΔAcvr1-*

*R206H* injected embryos (*Figure 4l,m,n*). These data demonstrate that Acvr1 requires a ligand-binding domain to respond to both Activin A and Bmp7, and suggest that Acvr1-R206H can activate pSmad1/5 signaling independently of both of these ligands. These data further support that FOP-ACVR1 mutant receptors have both ligand-independent and ligand-responsive activity.

We next evaluated the ability of Acvr1-R206H to activate pSmad2 signaling. A nuclear pSmad2 gradient forms along the margin of the zebrafish embryo during early gastrula stages in response to Nodal-Vg1 signaling and specifies endodermal and mesodermal cell fates (*Hill, 2018*; *Pelliccia et al., 2017*; *Montague and Schier, 2017*; *van Boxtel et al., 2015*; *Figure 4o*). Activin A can rescue loss of Nodal signaling by activating pSmad2 through Nodal receptors (*Gritsman et al., 1999*). Acvr1-R206H has been shown to signal in response to either BMP or Activin A ligand in cell culture and over-activate pSmad1/5, but not pSmad2, signaling (*Hatsell et al., 2015*; *Hino et al., 2015*; *Hildebrand et al., 2017*). Expression of Activin A alone in the developing zebrafish over-activates pSmad2 throughout the entire embryo (*Figure 4p*).

To test if Acvr1-R206H enhances signaling through the pSmad2 pathway, we injected Acvr1-R206H mRNA into *bmp7-/-*, Acvr1l-KD embryos with or without *Activin A* mRNA and measured pSmad2 immunofluorescence intensity. Embryos mutant for *bmp7* were used to eliminate possible competition between Activin A and Bmp7 for Acvr1 and type II BMP receptors and enhance any potential results. Expression of Activin A significantly increased mean pSmad2 signaling regardless of the presence of endogenous Acvr1l (*Figure 4p,q,t*). Acvr1-R206H alone did not increase the mean pSmad2 intensity compared to uninjected embryos and had significantly lower mean pSmad2 intensity compared to embryos injected with *Activin A* (*Figure 4r,t*). Activin A significantly increased mean pSmad2 intensity in Acvr1-R206H injected embryos, but no more than in the absence of Acvr1-R206H (*Figure 4s,t*). These data show that Activin A enhances pSmad2 signaling in the embryo, but Acvr1-R206H does not, and confirm that Acvr1-R206H over-activates pSmad1/5, but not pSmad2 signaling.

## ACVR1-R206H signals in the absence of the type I Bmpr1 receptor

The ability of FOP-ACVR1 to signal independently of ligand suggests the possibility that the mutant receptor could signal in the absence of receptor complex partners as well. In the developing zebrafish, both type I BMP receptors, Acvr1l and Bmpr1, are required for signaling and to pattern the embryo (*Mintzer et al., 2001*; *Little and Mullins, 2009*). BMP ligand is required for Acvr1l to associate with Bmpr1 in the zebrafish gastrula embryo (*Little and Mullins, 2009*). Previous studies showed that ACVR1-R206H retained the ability to over-activate BMP signaling when either BMPR1A or BMPR1B was knocked down in cell culture (*Hino et al., 2015*). However, given that these two genes have largely redundant activity (*Yoon et al., 2005*; *Wine-Lee et al., 2004*), one copy of BMPR1 (A or B) may be sufficient to allow ACVR1-R206H to signal.

Zebrafish have two *bmpr1a* genes (aa and ab) and two *bmpr1b* genes (ba and bb). To test if ACVR1-R206H can signal in the absence of all other type I BMP receptors, we first intercrossed *bmpr1aa+/-*; *bmpr1ab-/-* zebrafish (*Figure 5a*) to deplete *bmpr1a* gene function. We next injected the *bmpr1a*-deficient embryos with MOs against *bmpr1ba*, *bmpr1bb*, and *acvr1l* (designated as Type I KD fish henceforth). We then injected these Type I KD fish with human *ACVR1-R206H* mRNA, collected embryos at an early gastrula stage for pSmad1/5 immunostaining or at 12 to 30 hpf for phenotyping, analyzed them blindly, then genotyped for the *bmpr1aa* mutation. Using immunostaining, we confirmed that Flag-tagged ACVR1-R206H is expressed and localized to the cell membrane in *bmpr1a+/-* embryos (*Figure 5b*).

Both *bmpr1aa+/+*; *ab-/-* (*bmpr1a+/+*) and *bmpr1aa+/-*; *ab-/-* (*bmpr1a+/-*) embryos developed normally (*Figure 5c,d,f* column 1; *Figure 5—figure supplement 1*), but embryos null for both *bmpr1aa* and *bmpr1ab* (*bmpr1a-/-*) were severely dorsalized to a C4 phenotype (*Figure 5e,f* column 7; *Figure 5—figure supplement 1*). *bmpr1b* KD does not affect DV patterning of *bmpr1a+/+* or *+/-* embryos (*Figure 5f* column 2; *Figure 5—figure supplement 1*), demonstrating that Bmpr1aa with Acvr1l is sufficient for signaling and patterning the zebrafish embryo. However, *bmpr1b* KD further dorsalized *bmpr1a-/-* embryos to a C5 phenotype (*Figure 5f* column 8; *Figure 5—figure supplement 1*). Embryos were severely dorsalized to a C5 phenotype by KD of *bmpr1b* and *acvr1l* (*Figure 5f* column 3; *Figure 5—figure supplement 1*) and Type I KD embryos were also dorsalized to a C5 phenotype (*Figure 5f* column 9; *Figure 5—figure supplement 1*). However, ACVR1-R206H was able to ventralize Type I KD embryos, indicating that it does not require its normal signaling

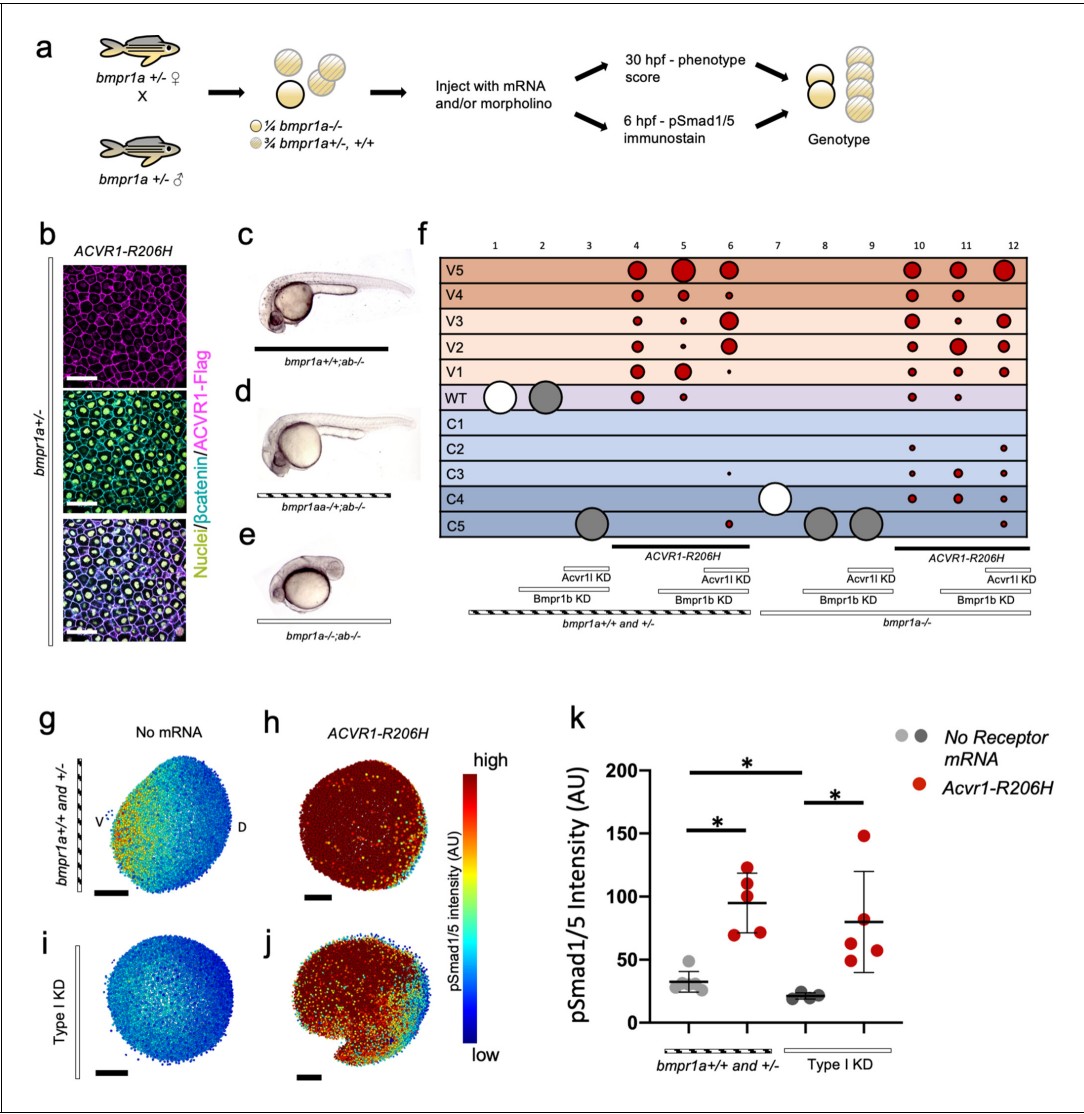

**Figure 5.** ACVR1-R206H signals in the absence of BMPR1. (a) Experimental workflow. *bmpr1aa+/-; ab-/-* fish were crossed with each other to generate *bmpr1a+/-* (*bmpr1aa+/+; ab-/-*), *bmpr1a+/-* (*bmpr1aa+/-; ab-/-*), and *bmpr1a-/-* (*bmpr1aa-/-;ab-/-*) embryos. One-cell stage eggs were injected with *bmpr1ba*, *bmpr1bb* morpholinos, and separately injected with an *acvr1l* morpholino to additionally knockdown these endogenous BMP receptors, and then were injected with *ACVR1-R206H* mRNA. Early gastrula (shield-65% epiboly) stage embryos were collected and fixed for immunostaining. At 12 to 30 hpf the remaining embryos were scored for DV patterning phenotypes. Following blindly immunostaining and imaging, or phenotyping, embryos were individually genotyped. (b) Representative immunostained embryo for ACVR1-Flag (magenta), β-Catenin (marking the cell membrane cyan), and nuclei (yellow) (N=5). (c-e) Representative 24 hpf phenotypes of (b) *bmpr1aa+/+; ab-/-*, (c) *bmpr1aa+/-; ab-/-*, and (d) *bmpr1aa-/-; ab-/-* embryos. (f) Injected embryo phenotypes at 12 to 30 hpf. Three pooled experiments. Columns: 1, N=104; 2, N=41; 3, N=69; 4, N=69; 5, N=78; 6, N=70; 7, N=36; 8, N=19; 9, N=25; 10, N=29; 11, N=23; 12, N=22. (g-j) Animal pole view of relative pSmad1/5 intensities (AU) within each nucleus in representative early-gastrula embryos. (d and f) Embryos are oriented with the ventral (V) side to the left and the dorsal (D) side to the right. (e and g) Dorsal side of the embryo could not be identified due to loss of the shield structure with ventralization. (g) *bmpr1a+/-* embryo (N=5). (h) *bmpr1a+/-* embryo injected with *ACVR1-R206H* mRNA (N=5). (i) *bmpr1a-/-* embryo with *acvr1l* and *bmpr1b* KD (Type I KD embryo) (N=4). (j) Type I KD embryo injected with *ACVR1-R206H* mRNA (N=5). (k) Mean nuclear pSmad1/5 fluorescence of injected embryos. Each dot represents the mean fluorescence of an individual embryo. Mean and standard deviation of each condition (e-h) is shown. * indicates P<0.05, ns indicates no significance.

The online version of this article includes the following source data and figure supplement(s) for figure 5:

**Source data 1.** Injected embryo phenotype raw numbers for *Figure 5f*.
**Source data 2.** Average nuclear pSmad1/5 fluorescence for *Figure 5k*.
**Source data 3.** Raw nuclear Smad1/5 flourescence for *Figure 5k*.
**Figure supplement 1.** ACVR1-R206H signals in the absence of all other type I BMP receptors.
**Figure supplement 2.** ACVR1-G328R signals in the absence of all other type I BMP receptors.
**Figure supplement 2—source data 1.** Injected embryo raw numbers for *Figure 5—figure supplement 2*.

partner, Bmpr1, to pattern the embryo (*Figure 5f* column 12; *Figure 5—figure supplement 1*). Similarly, ACVR1-G328R could ventralize Type I KD embryos, demonstrating it shares the ability to signal without Bmpr1 (*Figure 5—figure supplement 2*).

We examined how the pSmad1/5 gradient was affected in the injected embryos. ACVR1-R206H significantly increased the mean pSmad1/5 intensity compared to uninjected *bmpr1a+/-sibling* embryos (*Figure 5g,h,k*). Type I KD embryos lacked a pSmad1/5 gradient and had a significantly lower mean pSmad1/5 fluorescence than uninjected siblings (*Figure 5i,k*), demonstrating that loss of type I receptors results in loss of pSmad1/5 signaling. However, ACVR1-R206H significantly increased pSmad1/5 intensity in Type I KD embryos (*Figure 5j,k*), consistent with the observed ventralized phenotype. These data suggest that FOP-ACVR1 does not require BMPR1a or BMPR1b to phosphorylate Smad1/5 and that this mutant receptor does not require the presence of wild-type ACVR1-BMPR1 signaling complexes.

## ΔACVR1-R206H signals in the absence of the type I Bmpr1 receptor

To test if ligand-independent FOP-ACVR1 signaling requires Bmpr1 or endogenous Acvr1l, we injected Type I KD embryos with mouse *ΔAcvr1-R206H* or *Acvr1-R206H* and evaluated DV patterning phenotypes at 30 hpf and dorsal marker expression in 5- to 9-somite stage embryos. Interestingly, like *Acvr1-R206H* (*Figure 5*; *Figure 6a*, *Figure 6—figure supplement 1*), *ΔAcvr1-R206H* also ventralized Type I KD embryos (*Figure 6b* column 8; *Figure 6—figure supplement 1*), indicating that ligand-independent ACVR1-R206H signaling also does not require other type I BMP receptors to signal in patterning the zebrafish embryo.

We additionally evaluated expression of two dorsal markers, *pax2.1* and *krox20,* by whole-mount in situ hybridization. In wild-type 5- to 9-somite stage embryos, *pax2.1* is expressed in the midbrain-hindbrain (MHB) boundary (*Figure 6c*, white arrowhead at anterior) and *krox20* is expressed in rhombomeres 3 and 5 (*Figure 6c*, black arrowheads) (*Thisse et al., 2001*; *Strähle et al., 1993*; *Krauss et al., 1991*; *Hashiguchi and Mullins, 2013*). In *bmpr1a-/-* embryos, which develop to a C4 dorsalized phenotype, the MHB and rhombomere expression of *pax2.1* and *krox20*, respectively, were expanded laterally (*Figure 6d*). In Type I KD embryos, which display a C5 dorsalized phenotype, *pax2.1 and krox20* became radially expressed in the MHB and rhombomere 3 and 5 (*Figure 6e*). *Acvr1-R206H* mRNA injection rescued Type I KD embryos to a ventralized phenotype, characterized by dorsally restricted neural expression of *pax2.1* and *krox20* (*Figure 6f*) compared to Type I KD embryos. Injection of Type I KD embryos with *ΔAcvr1-R206H* mRNA, which lacks critical regions of the ligand-binding domain, similarly restricted expression of *pax2.1* and *krox20* to dorsal regions (*Figure 6g*). These data provide additional support that ligand-independent signaling by ACVR1-R206H also does not require Bmpr1a, Bmpr1b, or endogenous Acvr1l.

## ACVR1-R206H responds to ligand in the absence of Bmpr1

Since ligand facilitates association of Acvr1 and Bmpr1 in the developing zebrafish (*Little and Mullins, 2009*), we next examined if FOP-ACVR1 requires other type I BMP receptors for its enhanced ventralizing activity in response to ligand. We expressed human *ACVR1-R206H* in Type I KD embryos that were either deficient in BMP (by overexpressing the BMP ligand-binding inhibitor Chordin) or overexpressing Bmp7. Embryos were evaluated for DV patterning and dorsal marker gene expression.

Overexpression of Chordin dorsalized embryos in both the presence and absence of Bmpr1 (*Figure 7a* columns 1 and 9; *Figure 7—figure supplement 1*). While Bmp7 overexpression ventralized embryos in the presence of Bmpr1a (*Figure 7a* column 3; *Figure 7—figure supplement 1*), Bmp7 had no effect on patterning in Bmpr1a-deficient or Type I KD embryos, as expected (*Figure 7a* column 11 and 13; *Figure 7—figure supplement 1*). The inability of Bmp7 to affect Bmpr1a-deficient embryos, while it ventralizes Bmpr1a+/-embryos (*Figure 7a* columns 3,10,11), attests to the strong Bmpr1 loss of function. As in *Figure 5f*, ACVR1-R206H ventralized embryos even in the absence of all other type I BMP receptors (*Figure 7a*, column 15; *Figure 7—figure supplement 1*). Comparatively, overexpression of Chordin inhibited ventralization by ACVR1-R206H with or without Bmpr1 (*Figure 7a* columns 6 and 14, compare to columns 7 and 15; *Figure 7—figure supplement 1*), consistent with reduced BMP pathway activity. Overexpression of Bmp7

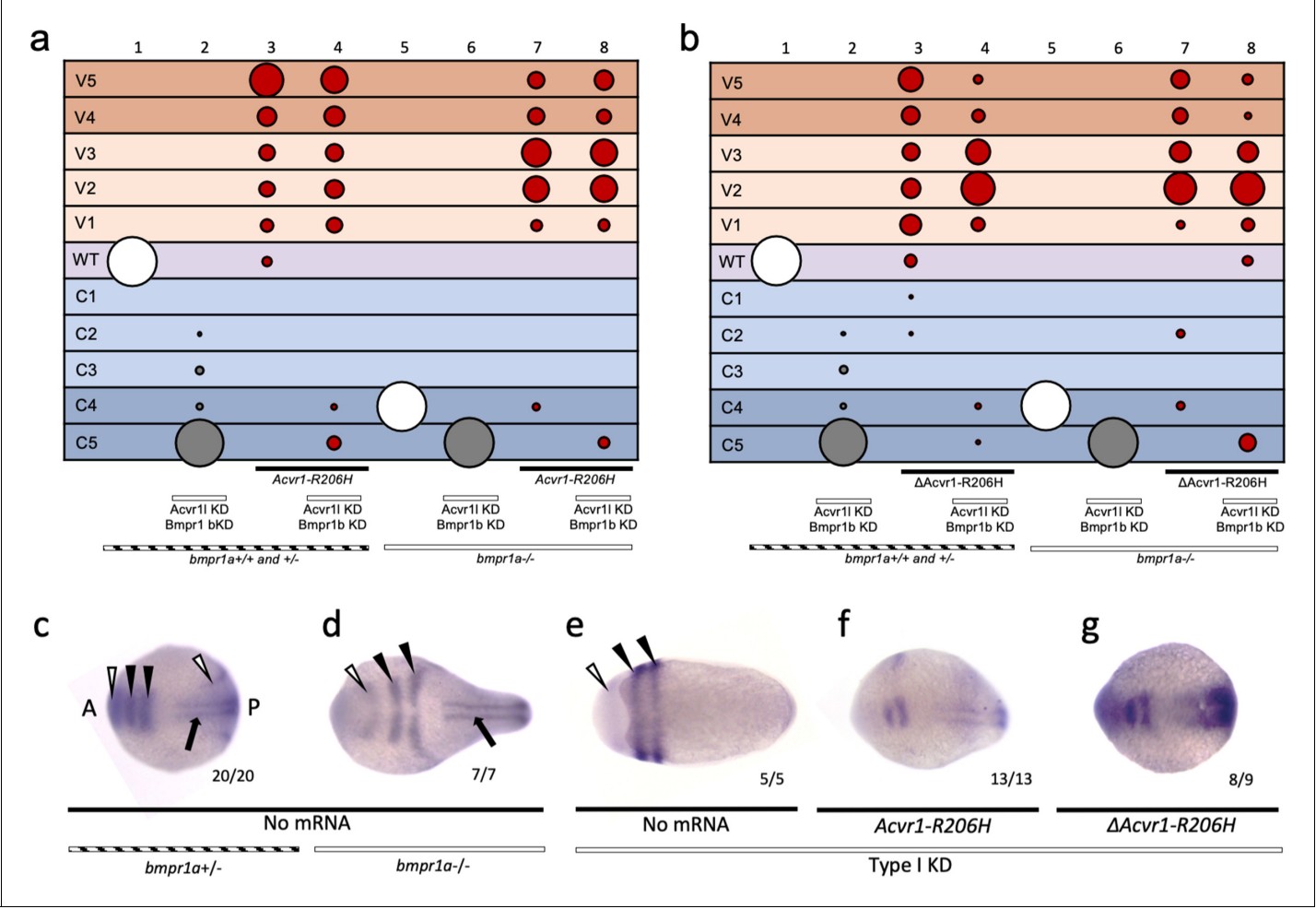

**Figure 6.** ΔACVR1-R206H signals in the absence of Bmpr1. (a–b) Injected *bmpr1a+/-* or *-/-* 12 to 30 hpf embryo phenotypes with *bmpr1b* KD, *acvr1* KD with or without *Acvr1-R206H* or *ΔAcvr1-R206H* mRNA. Two pooled experiments. (a) *Acvr1-R206H* injected embryos. Columns: 1, N = 150; 2, N = 80; 3, N = 98; 4, N = 89; 5, N = 76; 6, N = 36; 7, N = 31; 8, N = 30. (b) *ΔAcvr1-R206H* injected embryos. Columns: 1, N = 150; 2, N = 80; 3, N = 92; 4, N = 127; 5, N = 76; 6, N = 36; 7, N = 26; 8, N = 37. (c–i) Representative dorsal view of *pax2.1* (white arrowheads: anteriorly in the midbrain-hindbrain boundary and posteriorly in the pronephric mesoderm), *krox20* (black arrowheads; rhombomeres 3 and 5), and *myod* (black arrow; paraxial mesoderm) expression in 5–9 somite stage *bmpr1a+/-or -/-* embryos with *bmpr1b* KD, *acvr1* KD with or without *Acvr1-R206H* or *ΔAcvr1-R206H*. Two independent experiments. Embryos are oriented with the anterior (A) side left and the posterior (P) side right. Number of embryos that showed expression patterns similar to the representative embryos out of the total number of embryos analyzed is shown. (c) *bmpr1a+/-* embryo (d) *bmpr1a-/-* embryo (e) *bmpr1a-/-*embryo with *acvr1l* and *bmpr1b* KD (Type I KD embryo) (f) Type I KD embryo injected with *Acvr1-R206H* mRNA. (g) Type I KD embryo injected with *ΔAcvr1-R206H* mRNA.

The online version of this article includes the following source data and figure supplement(s) for figure 6:

**Source data 1.** Injected embryo raw numbers for *Figure 6a and b*.

**Figure supplement 1.** ΔACVR1-R206H signals in the absence of all other type I BMP receptors.

enhanced ventralization by ACVR1-R206H with or without Bmpr1, consistent with increased signaling (*Figure 7a*, compare columns 8 and 16, to columns 7 and 15; *Figure 7—figure supplement 1*).

We next evaluated 5–9 somite stage embryos for *pax2.1* and *krox20* expression using in situ hybridization. Compared to *bmpr1a+/+* and *+/-* embryos (*Figure 7b*, as in *Figure 6c*), *bmpr1a-/-* embryos displayed expanded neural expression of *pax2.1* and *krox20* (*Figure 7c*, as in *Figure 6d*) and Type I KD embryos had radialized expression of *pax2.1* and *krox 20* (*Figure 7d*, as in *Figure 6e*). Type I KD embryos co-injected with *ACVR1-R206H* and *chordin* mRNA had dorsally-restricted expression of *pax2.1* and *krox20* compared to Type I KD embryos (*Figure 7d,e*), although not as restricted as *bmpr1a+/-* embryos (*Figure 7b*), consistent with moderately dorsalized

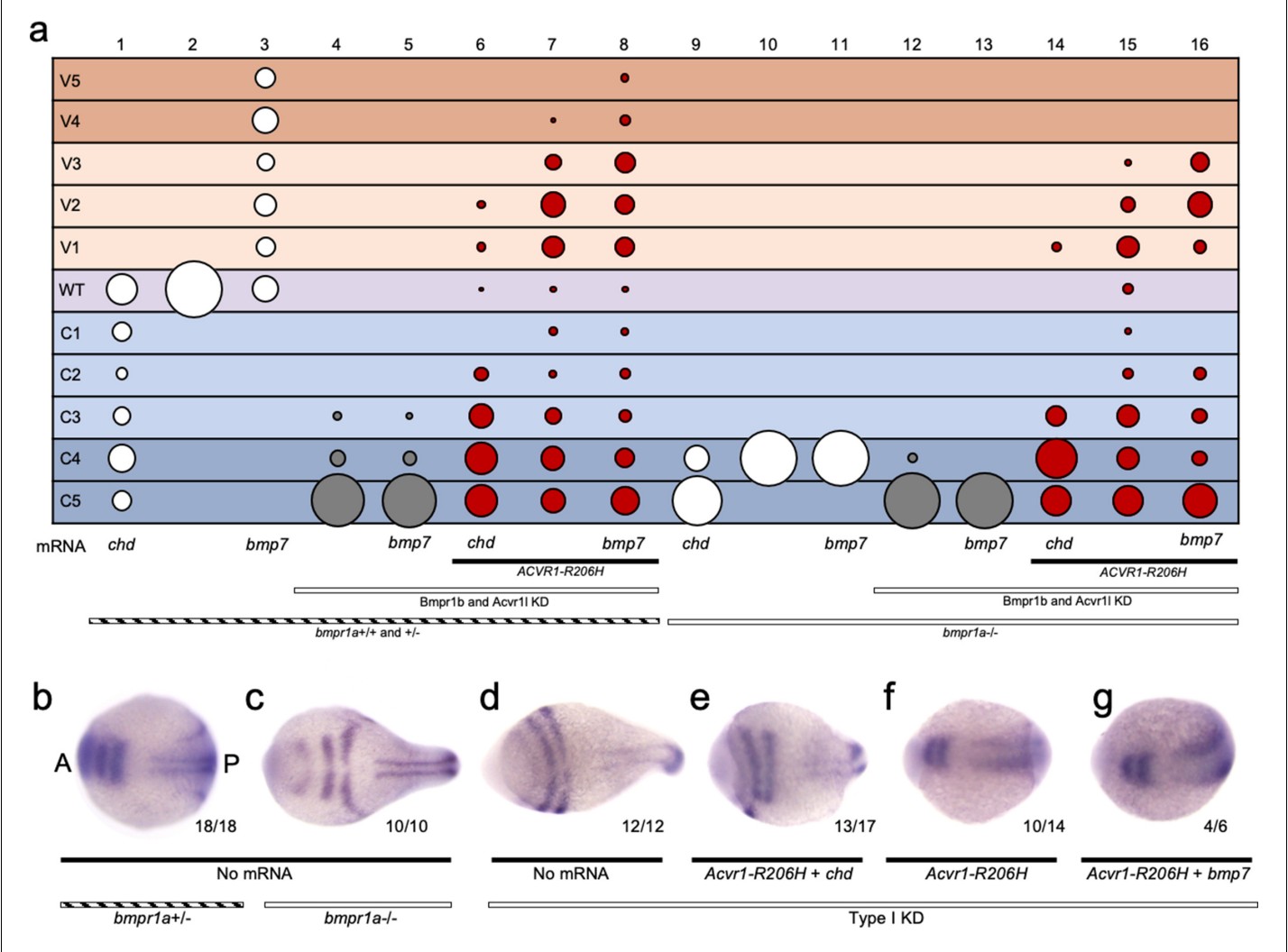

**Figure 7.** ACVR1-R206H responds to ligand in the absence of all other type I BMP receptors. (**a**) 12–30 hpf phenotypes of *bmpr1a+/-*or *-/-* embryos with *bmpr1b* KD and *acvr1l* KD with or without ACVR1-R206H, with or without *chordin* (*chd*) mRNA or *bmp7* mRNA. Four pooled experiments plus two experiments with controls only. Columns: 1, N = 172; 2, N = 160; 3, N = 106; 4, N = 135; 5, N = 104; 6, N = 121; 7, N = 169; 8, N = 117; 9, N = 32; 10, N = 91; 11, N = 17; 12, N = 57; 13, N = 29; 14, N = 28; 15, N = 49; 16, N = 49. (**b–g**) Representative dorsal view of *pax2.1*, *krox20*, and *myod* expression in 5–9 somite stage *bmpr1a+/-*or *-/-* embryos with *bmpr1b* KD, *acvr1* KD, with or without ACVR1-R206H, and with or without *chordin* or *bmp7* mRNA. Three experiments. Embryos are oriented with the anterior (**A**) side left and the posterior (**P**) side right. Number of embryos that showed expression patterns similar to the representative embryos out of the total number of embryos analyzed is shown. (**b**) *bmpr1a* +/- embryo (**c**) *bmpr1a-/-*embryo (**d**) *bmpr1a-/-*embryo with *acvr1l* and *bmpr1b* KD (Type I KD embryo) (**e**) Type I KD embryo injected with ACVR1-R206H and *chd* mRNA (**f**) Type I KD embryo injected with ACVR1-R206H mRNA (**g**) Type I KD embryo injected with ACVR1-R206H and *bmp7* mRNA.

The online version of this article includes the following source data and figure supplement(s) for figure 7:

**Source data 1.** Injected embryo raw numbers for *Figure 7a*.

**Figure supplement 1.** ACVR1-R206H responds to ligand in the absence of all other type I BMP receptors Representative 12 and 30 hpf phenotypes of injected embryos.

phenotypes. Type I KD embryos injected with *ACVR1-R206H* or co-injected with *ACVR1-R206H* and *bmp7*, expressed *pax2.1* and *krox20* in patterns similar to, or more dorsally-restricted than *bmpr1a +/-* embryos, consistent with rescue to wild-type or ventralized phenotypes (*Figure 7f and g*). These results suggest that ACVR1-R206H can respond to ligand in the absence of Bmpr1a, Bmpr1b, and endogenous Acvr1l.

## Discussion

Using an in vivo zebrafish model, a highly informative and sensitive vertebrate system for BMP signaling activity during DV patterning, we investigated the signaling mechanism of the ACVR1 type I BMP receptor and the effects of ACVR1 activating mutations that cause a rare genetic disorder of ectopic bone formation. We show that although wild-type ACVR1 requires its ligand-binding domain for signaling, the FOP-ACVR1 mutants, ACVR1-R206H and ACVR1-G328R, do not require a ligand-binding domain to over-activate pSmad1/5 signaling. However, these mutant receptors do retain the ability to respond to BMP and Activin A ligand when their ligand-binding domain is intact. We further show for the first time that Bmpr1, which is a required partner for signaling by WT Acvr1l, is dispensable for both ligand-independent and ligand-responsive signaling by mutant FOP-ACVR1. Our data support that the causative mutations of FOP allow the mutant receptor to bypass normal ligand-receptor complex assembly. This aberrant signaling highlights the importance of ligand-receptor complexes to facilitate and regulate signaling by ACVR1. Elucidation of FOP-ACVR1 signaling mechanisms not only provides insight into therapeutic targets for treating FOP, but also gives us unique insight into how BMP signaling is regulated.

Recent studies have shown that FOP-ACVR1 has acquired the ability to signal through pSmad1/5 in response to the TGFβ family ligand Activin A, which normally binds ACVR1b to signal through pSmad2/3 (*Hatsell et al., 2015*; *Lees-Shepard et al., 2018*; *Hino et al., 2015*). Our study confirms that ACVR1-R206H is able to respond to both its normal activating ligand, Bmp7, and to Activin A, in an in vivo model, demonstrating that FOP-mutant ACVR1 has an expanded repertoire of ligands to which it can respond. These data support a mechanism of over-active signaling induced by availability of multiple ligands for ACVR1-R206H. Interestingly, the magnitude of Bmp7 response by ACVR1-G328R appeared to be greater than that of ACVR1-R206H (*Figure 3c,d*). While multiple factors could be contributing to this observation, other studies have shown similar trends (*Haupt et al., 2018*), suggesting that the signaling mechanisms of the two mutants are not identical. An interesting next experiment would be to compare Activin A responses of ACVR1-R206H and -G328R in vivo.

In this study, we also demonstrate in an in vivo vertebrate system that neither Acvr1-R206H nor Acvr1-G328R require a ligand-binding domain to signal. Further supporting ligand-independent pathway activation, we showed that Acvr1-R206H and Acvr1-G328R are able to pattern the zebrafish embryo in the absence of their normal obligatory ligand, Bmp2/7 (*Figure 3*). This does not preclude the possibility that other components of an FOP-ACVR1 signaling complex play a role in binding ligands other than BMP. We show that Acvr1-R206H pSmad1/5 signaling also is inappropriately activated by Activin A in the zebrafish. By RNAseq and wholemount in situ hybridization, low levels of *activin A* expression are detectable in the prechordal plate during late gastrulation (75% epiboly) (*Thisse et al., 2004*; *White et al., 2017*), a stage later than our analysis of pSmad1/5 at early gastrulation (shield to 65% epiboly) (*Figures 2* and *4d,e*). However, another report detected low *activin A* expression during early gastrulation by RT-PCR (*Hashiguchi et al., 2008*), thus we cannot exclude Activin A as a factor in FOP-ACVR1 signaling in *Figure 3*. We show, however, that ligand-binding deficient ACVR1 does not respond to Bmp7 or Activin A ligand (*Figure 4k–n*), demonstrating that the ligand-binding domain of ACVR1-R206H is required for ligand response and FOP-ACVR1 does not require ligand to signal (*Hildebrand et al., 2017*).

Studies suggest that not only is ligand critical for normal type I receptor complex assembly, but ligand also mediates which receptors assemble together through differential ligand-receptor affinity (*Yadin et al., 2016*; *Little and Mullins, 2009*; *Heinecke et al., 2009*; *Allendorph et al., 2007*; *Antebi et al., 2017*). In zebrafish, Acvr1l and its type I BMP receptor partner Bmpr1 will not associate with each other and signal in the absence of appropriate ligands (*Little and Mullins, 2009*). Therefore, the ability of ACVR1-R206H and -G328R to signal in response to novel ligands suggests an acquired ability to signal in novel receptor complex combinations. Conversely, the ability of the mutant receptors to signal in the absence of an intact ligand-binding domain suggests an acquired ability to signal in the absence of normally-regulated complex formation. In the context of the developing zebrafish, these mutations decouple the activity of ACVR1 from its morphogen signal, Bmp2/7. Indeed, in this study, we also determined that ACVR1-R206H does not require its normal type I BMP receptor partner, Bmpr1, for its signaling activity.

We note that in our experimental system, residual maternal *bmpr1aa* mRNA remains in our *bmpr1aa-/-, ab-/-* mutants; however, the inability of Bmp7 overexpression to even partially rescue

the C4 dorsalized phenotype of *bmpr1aa-/-; ab-/-* mutants or the C5 dorsalization of Type I KD mutants indicates that residual *bmpr1aa+/-* maternal transcript has little or no effect on DV patterning (*Figure 5f*). Further, the inability of Bmp7 overexpression to rescue Type I KD embryos suggests that any remaining receptor from our knockdown was not sufficient to contribute to DV patterning, indicating a severe loss of Type I receptor function.

The signaling activity that we observed in our ligand-binding mutant ACVR1 receptors may represent a baseline constitutive activity by ACVR1-R206H and -G328R in the absence of receptor complex assembly. All confirmed ACVR1 mutations in FOP are located within the GS or protein kinase domains and are theorized to alter receptor phosphorylation and, as a result, the activation energy required for ACVR1 to signal (*Chaikuad et al., 2012*; *Shore and Kaplan, 2010*). Previous studies have suggested that the FOP mutations weaken binding by the type I receptor inhibitor FKBP12, allowing for constitutive or 'leaky' signaling (*Chen et al., 1997*; *Huse et al., 1999*; *Groppe et al., 2011*). However, more recent evidence suggests that this loss of FKBP12 inhibition does not fully account for the level of increased activity displayed by the FOP mutant receptors (*Machiya et al., 2018*). In addition, ACVR1-R206H has been shown to require type II BMP receptors and GS domain activation to signal (*Hino et al., 2015*; *Bagarova et al., 2013*; *Le and Wharton, 2012*). In the absence of ligand-induced complex assembly with other receptors, GS domain activation of FOP-ACVR1 may occur by stochastic association with free type II BMP receptors. These data highlight that even more remains to be elucidated about the mechanism by which BMP receptor components assemble and coordinately regulate signaling.

Conversely, the Bmp7 and Activin A ligand-responsive activity that we observed by ACVR1-R206H and -G328R may reflect mutation-induced stabilization of signaling activity by ligand-facilitated receptor complex assembly. Under normal circumstances, WT Acvr1l is an obligate type I receptor partner with either Bmpr1a or Bmpr1b (*Little and Mullins, 2009*). Our results show that FOP-ACVR1 can respond to Bmp7 ligand without its normal type I BMP receptor partner, Bmpr1. These data suggest that ACVR1-R206H is either able to respond to ligand on its own or with novel receptor partners. An important future experiment would be to test if ACVR1-R206H requires Bmpr1 or other type I receptors to respond to Activin A.

In the absence of Bmpr1, ACVR1-R206H may homodimerize with itself to signal. BMPR1, although not ACVR1, has been shown to form inert pre-formed homodimer complexes that are poised to respond to ligand (*Marom et al., 2011*). ACVR1, however, normally binds ligand poorly on its own, indicating that this type I receptor may not easily associate or form stable complexes with ligand in the absence of other receptor partners (*Heinecke et al., 2009*). We also note that the type I BMP receptor, ACVRL1 (ALK1), was not targeted for knockdown in this study; however, expression studies show that ACVRL1 is not detectable by RNAseq until 75% epiboly, after the stage in which we see expanded pSmad1/5 signaling, and it is not detectable by in situ during the first 24 hr of development (*Thisse et al., 2004*; *White et al., 2017*). Our study, however, cannot exclude that low levels of ACVRL1 play a role in the ability of ACVR1-R206H to respond to Bmp7 in the absence of BMPR1. In our study, the type II BMP receptors are also present in the zebrafish embryo and may be sufficient to form a working signaling complex with FOP-ACVR1 as the only type I receptor and still respond to ligand. Previous reports have shown that both BMPR2, a type II receptor with high affinity for BMP ligand, and ACVR2a, a type II receptor with high affinity for Activin A, are required for signaling by ACVR1-R206H (*Yadin et al., 2016*; *Hino et al., 2015*; *Walton et al., 2012*), and an important next step would be to test this requirement in our in vivo system.

Recent studies have shown that TGFβR1 can associate with ACVR1 and initiate pSmad1/5 signaling (*Ramachandran et al., 2018*). TGFβR1 normally signals with TGFβR2 in response to TGFβ ligand, components with which ACVR1 is not known to associate. ACVR1b, the Nodal/Activin receptor, is also present in the zebrafish embryo. Notably, ACVR1b uses the same type II receptors, ACVR2a and ACVR2b, as ACVR1 to signal, although these complexes normally signal through pSmad2/3 rather than pSmad1/5. The ability of FOP-ACVR1 to signal in response to Activin A, may occur by signaling through components of the normal Activin signaling complex: ACVR1b and ACVR2. Our data show that if ACVR1 signals by complexing with Acvr1b, this was not sufficient to sequester Acvr1b and reduce pSmad2 signaling in our system (*Figure 4t*). We show that there is no significant difference in pSmad2 intensity in embryos injected with Activin A alone or with Acvr1-R206H. Much remains to be investigated to elucidate the mechanism of an ActivinA/ACVR1-R206H response.

The pathological significance of the ACVR1-R206H signaling modalities elucidated in this study remain to be further investigated. One possibility is that they represent two proposed modes of HO initiation in FOP patients: spontaneous and injury-induced. Ligand-independent basal activity of FOP-ACVR1 may lead to increasing levels of signaling within tissues and the initiation of spontaneous HO when a threshold is reached with no obvious acute inciting factor (*Shore and Kaplan, 2010*). Ligand-responsive hyperactivity may account for injury-induced HO, in which cascades of signaling in response to injury-induced ligands, including Activin A (*Hatsell et al., 2015*; *Lees-Shepard et al., 2018*; *Hino et al., 2015*), contribute to increased activity by FOP-ACVR1. It is important to note, however, that these two modes of HO initiation may not represent separate mechanisms, and ligand independent and responsive activation may play more nuanced roles in the pathogenesis of disease. Research to identify FOP treatment strategies has examined multiple drug targets in ACVR1 signaling including ligand binding and ACVR1 kinase activity (*Cappato et al., 2018*). Our data suggest that directly targeting the kinase activity of FOP-ACVR1 to inhibit both ligand independent and responsive signaling by the receptor would be a highly promising strategy for treatment.

# Materials and methods

**Key resources table**

| Reagent type (species) or resource | Designation | Source or reference | Identifiers | Additional information |
|---|---|---|---|---|
| Genetic reagent (*Danio rerio*) | bmpr1aa$^{p3/+}$ | This paper | p3 | |
| Genetic reagent (*Danio rerio*) | bmpr1ab$^{sa0028}$ | ZIRC | sa0028 | |
| Genetic reagent (*Danio rerio*) | bmp7a$^{sb1aub}$ | *Schmid et al., 2000* | sb1aub, RRID:ZFIN_ZDB-GENO -100415-21 | |
| Sequence-based reagent | Alk 8 (Acvr1l) Morpholino 4 | Gene Tools LLC | MO4-acvr1l (previously MO2-acvr1l) | TGCCTTTCAGTATT CGCACAGCCAG |
| Sequence-based reagent | Alk8 (Acvr1l) Morpholino 2 | Gene Tools LLC | MO2-acvr1l (previously MO3-acvr1l) | GATTCATGTTTGTG TTCAATTTCCG |
| Sequence-based reagent | Alk6a (Bmpr1ba) Morpholino 1 | Gene Tools LLC | MO1-bmpr1ba | AGAACTCCAGTGAG CCAGAGAATCC |
| Sequence-based reagent | Alk6b (Bmpr1bb) Morpholino 1 | Gene Tools LLC | MO1-bmpr1bb | ACTGCTCCACAGCT ACTCCACACTG |
| Recombinant DNA reagent | Human ActivinA (plasmid) | Origene | Cat #RC203226 INHBA, inhibinβa, ActivinA | Cloned into pCS2 backbone |
| Recombinant DNA reagent | Zebrafish *bmp7a* (plasmid) | *Schmid et al., 2000* | bmp7a | Cloned into pCS2 backbone |
| Recombinant DNA reagent | Zebrafish *chordin* (plasmid) | *Miller-Bertoglio et al., 1997* | chd | Cloned into pCS2 backbone |
| Recombinant DNA reagent | Human *ACVR1* (plasmid) | The Shore Lab. This Paper. *Shen et al., 2009* | ACVR1 | Cloned into pCS2 backbone |
| Recombinant DNA reagent | Human *ACVR1-R206H* (plasmid) | The Shore Lab. *Shen et al., 2009* | ACVR1-R206H | Cloned into pCS2 backbone |
| Recombinant DNA reagent | Human *ACVR1-G328R* (plasmid) | The Shore Lab. This Paper. (*Shore et al., 2006*). | ACVR1-G328R | Cloned into pCS2 backbone |
| Recombinant DNA reagent | Mouse *Acvr1* (plasmid) | *Haupt et al., 2014* | | Cloned into pCS2 backbone |

*Continued on next page*

*Continued*

| Reagent type (species) or resource | Designation | Source or reference | Identifiers | Additional information |
|---|---|---|---|---|
| Recombinant DNA reagent | Mouse *Acvr1-R206H* (plasmid) | *Haupt et al., 2014* | p.R206H | Cloned into pCS2 backbone |
| Recombinant DNA reagent | Mouse *Acvr1-G328R* (plasmid) | *Haupt et al., 2014* | p.G328R | Cloned into pCS2 backbone |
| Recombinant DNA reagent | Mouse Δ*Acvr1* (plasmid) | *Haupt et al., 2014* | ΔLBD-Acvr1 | Cloned into pCS2 backbone |
| Recombinant DNA reagent | Mouse Δ*Acvr1-R206H* (plasmid) | *Haupt et al., 2014* | ΔLBD-Acvr1 | Cloned into pCS2 backbone |
| Recombinant DNA reagent | Mouse Δ*Acvr1-G328R* (plasmid) | *Haupt et al., 2014* | ΔLBD-Acvr1 | Cloned into pCS2 backbone |
| Antibody | Anti-pSmad1/5/8 (rabbit monoclonal) | Cell Signaling | Cat #13820 RRID:AB_2493181 | IF (1:200) |
| Antibody | Anti-pSmad2/3 (rabbit monoclonal) | Cell Signaling | Cat #8828, RRID:AB_2631089 | IF (1:800) |
| Antibody | Anti-Beta catenin (mouse monoclonal) | Sigma | Cat #C7207, RRID:AB_476865 | IF (1:1000) |
| Antibody | Anti-HA (rabbit polyclonal) | Invitrogen | Cat #71–5500, RRID:AB_87935 | IF (1:500) |
| Antibody | Anti-Flag (rabbit polyclonal) | Sigma | Cat #F7425, RRID:AB_439687 | IF (1:500) |
| Antibody | Anti-rabbit Alexa 647 (goat polyclonal) | Invitrogen | Cat #A-21245, RRID:AB_2535813 | IF (1:500) |
| Antibody | Anti-mouse Alexa 594 (goat polyclonal) | Molecular Probes | Cat #A21123, RRID:AB_141592 | IF (1:500) |
| Other | Sytox green | Fisher | Cat #S7020 | IF (1:2000) |
| Sequence-based reagent | *pax2.1* zebrafish in situ probe | *Krauss et al., 1992* | | |
| Sequence-based reagent | *krox20* zebrafish in situ probe | *Oxtoby and Jowett, 1993* | | |
| Sequence-based reagent | *myod* zebrafish in situ probe | *Weinberg et al., 1996* | | |
| Commercial assay, Kit | mMESSAGE mMACHINE SP6 Transcription Kit | ThermoFisher | Cat #AM1340 | |
| Software, algorithm | Fiji (ImageJ) | Fiji | | https://fiji.sc/#download |
| Software, algorithm | Imaris | Oxford Instruments | Imaris 9.6 | https://imaris.oxinst.com/ |

## Zebrafish

Procedures involving animals were approved by the University of Pennsylvania IACUC. Adult zebrafish were kept at 28°C in a 13 hr light/11 hr dark cycle. All zebrafish husbandry was performed in accordance with institutional and national ethical and animal welfare guidelines. The characterized mutant *bmp7a*^sb1aub (bmp7-/-) (*Schmid et al., 2000*) and *bmpr1aa*^p3/+; *bmpr1ab* ^sa0028 (bmpr1aa+/-; bmpr1ab-/-) were used in this study. *bmp7*^sb1aub fish were maintained as homozygous mutant stocks by rescuing the C5 dorsalized embryonic phenotype with *bmp7* mRNA injection. *bmpr1aa*+/-; *bmpr1ab*-/-fish were generated by intercrossing and genotyping for the *bmpr1aa* allele (described below). Embryos used in these experiments were 0–48 hpf. Embryos were maintained at 28–32°C in E3 solution. Sex/gender was not accounted for as zebrafish sex determination takes place later, during juvenile stages of development (*Santos et al., 2017*).

## CRISPR generation and identification of *bmpr1aa* mutant allele

A mutant allele of *bmpr1aa* (*bmpr1aa[P3]*) in the zebrafish was generated using CRISPR-Cas9 mutagenesis. The target site GGTATAAGTGGCAGACAGAG in exon 8 (out of 13) in the kinase domain, was selected with the assistance of the web tool CHOPCHOP (https://chopchop.cbu.uib.no). Single guide RNAs (sgRNAs) were designed to utilize the T7 Promoter. sgRNAs were synthesized in-vitro from PCR amplified templates using the cloning-free method of *Gagnon et al., 2014* with the following changes: sgRNA templates were amplified using Phusion Polymerase 40 uL reaction (Thermo-Fischer F553S). sgRNA templates were purified after amplification using the MinElute purification kit (QIAGEN 28004). sgRNAs were synthesized from the templates in vitro using the MEGAshortscript T7 kit (ThermoFischer AM1354). Megashortscript reactions were run overnight rather than the recommended 2–4 hr, as this was found to increase yield. sgRNA was purified post synthesis using the ethanol precipitation protocol from *Gagnon et al., 2014*. Purified sgRNAs were assessed and quantified visually by running a dilution series on a glyoxal/sodium phosphate buffer RNA gel and comparing to RiboRuler (ThermoFischer SM1821), as the Nanodrop was found to be an unreliable to quantify these guide RNAs.

Immediately prior to injection, 2.5 µL of undiluted sgRNA were mixed with 1 µl of 5 mg/mL Cas9-NLS protein (PNA Bio CP01-50), 1 µl of Phenol Red, and 0.5 µl 1M KCl (total volume 5 µl), this solution was kept at room temperature for the duration of the injection. This mixture was injected in volumes of 2 nl into wild type and *bmpr1ab[sa0028]* embryos.

CRISPR efficiency was assessed using High-Resolution Melt Analysis (HRMA) (*Dahlem et al., 2012*). At 48 hpf, a subset of the injected embryos was sacrificed for HRMA. HRMA was performed using the MeltDoctor HRM master mix (ThermoFischer 4415440). Embryos from injections with greater than 50% efficiency were raised to adulthood, and outcrossed to WT or *bmpr1ab-/-* fish. 12 embryos from each outcross were sacrificed for Sanger Sequencing. DNA was isolated from lysed embryos and the target sequence was amplified with PCR. Products from these PCR reactions were purified using the Qiaquick PCR purification kit (Qiagen 28106). Purified PCR products were quantified via Nanodrop and sequenced by the University of Pennsylvania Sequencing core by Sanger sequencing. Mutant alleles were identified by directly reading trace data using Lasergene SeqMan Pro.

A mutant allele with a 53 base pair deletion and a 29 base pair insertion, which includes the exon eight splice acceptor site was identified. Siblings of allele-carrying embryos were raised to adulthood, genotyped, and in-crossed. In the *bmpr1ab-/-* background, homozygous mutants display a C4 phenotype. Mutant embryos were re-sequenced to confirm the allele sequence. RNA was later isolated from homozygous mutant embryos at 12 hpf, a stage at which they could be identified phenotypically, using Trizol (ThermoFischer 15596026), and reverse transcribed to create cDNA using the SuperScript II kit (ThermoFischer 18064014). The *bmpr1aa* transcript was amplified by PCR, sequenced and compared to WT transcript. Sequence analysis revealed that the mutation lead to the inclusion of the intron between exons 7 and 8, which contains a premature stop codon.

CRISPR template synthesis oligos:

> Gene specific Oligo: TAATACGACTCACTATAGGTATAAGTGGCAGACAGAGGTTTTAGAGCTAGAAATAGCAAG
> Constant Oligo: AAAGCACCGACTCGGTGCCACTTTTTCAAGTTGATAACGGACTAGCCTTATTTTAACTTGCTATTTCTAGCTCTAAAAC
> HRMA primers
> Forward: AGAGGAGTCAGGAGTGATCTCTTT
> Reverse: TTGATGAGGTCTTTCAGGGATT

Target site Sequencing Primers:

> Amplification Forward Primer: CCTGTTTTTCCACATCACTGAA
> Amplification Reverse Primer: TAGAGGCTCCTGTGCCATTTAT
> Sequencing Primer: TCTATATTTTTGCCTGGCCCTA cDNA Sequencing Forward Primer: CAAGACAATTTGACAATGCGTCA cDNA Sequencing Reverse Primer: TCAGATTTTAATGTCTTGAGATTCCACC

Mutant Allele Sequence:

Wildtype sequence (deleted bases underlined): ATAAGAGGAGTCAGGAGTGATCTC
TTTAACATCAAGGATACNAAAAAAACAGCTTTGACTGTGTTTTGTCATCAGGTATAAG
TGGCAGACAGAGAGGCAGCGCTACCACAGAGACCTGGAGCAAGACGAGGCCTTTA
TCCCAGCAGGAGAATCCCTGAAAGA
Mutant sequence (inserted bases in bold):
ATAAGAGGAGTCAGGAGTGATCTCTTTAACATCAAGGATA**GTCCGTTATCAAC
TTGAAAAAGTGGCACC**GAGGCAGCGCTACCACAGAGACCTGGAGCAAGACGAGGCC
TTTATCCCAGCAGGAGAATCCCTGAAAGA

## Genotyping

Adult and embryonic genomic DNA was obtained using HotShot DNA isolation. Genotyping of adults and embryos for *bmpr1aa* was performed using KASPar genotyping (*Smith and Maughan, 2015*). Primers were designed and synthesized by LGC Bioscience Technologies. The following sequence was submitted for primer design: ATAAGAGGAGTCAGGAGTGATCTCTTTAACA TCAAGGATA[CNAAAAAAACAGCTTTGACTGTGTTTTGTCATCAGGTATAAGTGGCAGACAGA/G TCCGTTATCAACTTGAAAAAGTGGCACC]GAGGCAGCGCTACCACAGAGACCTGGAGCAAGAC-GAGGCCTTTATCCCAGCAGGAGAATCCCTGAAAGA (Primer sequences are proprietary, LGC Bio-search technologies). Immunostained embryos were recovered after photographing and placed in methanol prior to Hotshot and KASPar genotyping. Alternatively, *bmpr1aa* mutant fish were geno-typed by conventional PCR using Choice Taq Blue Mastermix (Denville CB4065-7), and the afore-mentioned HRM primers.

## mRNA synthesis

Human, mouse, or zebrafish *ACVR1* cDNAs were cloned into the pCS2+ expression vector. Mouse *Acvr1* constructs contained an HA tag inserted after amino acid M34. ΔAcvr1 constructs contained a 64 amino acid deletion (C35-C99) within the extracellular domain (*Haupt et al., 2014*). Human *ACVR1* constructs all contained a C-terminal Flag tag. mRNA was synthesized using SP6 mMessage machine kit (Sigma Aldrich) and purified using phenol:chloroform extraction. The mRNA was stored in nuclease-free water at −80°C.

## Microinjection of one-cell stage zebrafish embryos

Eggs at 0–15 min post fertilization were collected in E3 media and injected at 22°C. For each experiment, each mRNA was injected into eggs with the same calibrated needle. For serial injections, eggs were loaded onto the same plate and injected with the first mRNA or morpholino, a subset were then set aside for controls and the remainder were injected with the next mRNA or morpholino. This process was repeated multiple times in different orders to ensure consistency between experimental and control conditions. Injection concentrations of each mRNA were determined based on pheno-typic evaluation. In cases in which multiple mRNA syntheses were used, different mRNA concentrations were used to achieve the same phenotype due to presumed inconsistent mRNA 5' capping. Working concentrations of mRNA based on Nanodrop spectrometer measurements: 2.5–200 pg *mAcvr1* and Δ*mAcvr1* mRNAs, 65–250 pg *hACVR1* mRNAs, 1 ng *chordin* mRNA, 200 pg *bmp7* for rescue of *bmp7-/-* to a wildtype phenotype, 500 pg-1 ng of *bmp7* was used for overexpression experiments, 5–10 pg of Activin A mRNA. Morpholinos were synthesized by Gene Tools LLC and reconstituted in Daneaue solution at 25 mg/ml. A morpholino mixture of 2.3 ng Alk8MO4 (5'TGCC TTTCAGTATTCGCACAGCCAG3') and 9.2 ng Alk8MO2 (5'GATTCATGTTTGTGTTCAATTTCCG3') was used to knockdown endogenous *acvr1l*. To knock down all the type I BMP receptors, 17 ng Alk8MO2 was co-injected with a mixture of 5 ng Alk6aMO1 (5'AGAACTCCAGTGAGCCAGAGAA TCC3') and 2 ng Alk6bMO1 (5'ACTGCTCCACAGCTACTCCACACTG3'). All morpholinos bind inde-pendent sequences upstream of their target gene start sites to inhibit translation (*Little and Mullins, 2009*; *Bauer et al., 2001*).

## Phenotypic evaluation

Embryos between 12 to 48 hpf were categorized into dorsoventral patterning phenotypes (*Figure 1g*). All images of embryos were photographed in E3 media with a Leica IC80HD. Injection results and controls from multiple experiments were then pooled. Phenotype pictures were

corrected with background subtraction using ImageJ, and white balanced using Adobe Photoshop. Embryo bubble graphs were generated by pooling total numbers of embryos within each phenotypic category (C5-V5, *Figure 1g*) in each condition. These numbers were converted into percent of total embryos to generate the bubble graphs using excel bubble graph. (*Figure 1—source data 1*, *Figure 1—figure supplement 2—source data 1*, *Figure 3—source data 1*, *Figure 5—source data 1*, *Figure 5—figure supplement 2—source data 1*, *Figure 6—source data 1*, *Figure 7—source data 1*).

## Immunofluorescence

P-Smad1/5 immunostaining and imaging were performed as previously described (*Zinski et al., 2017*; *Zinski et al., 2019*). For all immunostaining, embryos were fixed in 4% formaldehyde in PBST between shield stage and 60% epiboly (approximately 6–7 hpf), blocked with 10% FBS in PBST, and probed overnight at 4°C with a 1:200 dilution of anti-PSmad1/5/9 (Cell Signaling 13820). Embryos were then treated overnight at 4°C with a 1:500 dilution of antibody Alexa 647 (Invitrogen A-21245) and 1:2000 of Sytox green (Fisher S7020) diluted in blocking solution. Stained embryos were stored in the dark at 4°C in PBST for up to 2 months.

P-Smad2 immunostaining was performed as described for P-Smad1/5 immunostaining with the following changes: Primary antibody used was anti-PSmad2/3 (1:800; Cell Signaling 8828), embryos were dehydrated in MeOH after fixing, rehydrated embryos were incubated in Acetone at −20C prior to blocking.

Receptor tag immunostaining was performed as described for P-Smad1/5. Primary antibodies used were: 1:200 of anti-HA (Invitrogen 71–5500) or 1:200 of anti-Flag (Sigma F7425), and 1:1000 of anti-beta-Catenin (Sigma C7207) diluted in blocking solution. Secondary antibodies used were: 1:500 of Alexa 546 (Molecular Probes A21123), and/or 1:2000 of Sytox green (Fisher S7020).

## Immunofluorescence imaging and analysis

Prior to imaging, immunostained embryos were gradually dehydrated in MeOH and then cleared using BABB: a 1:2 ratio of benzyl alcohol (Sigma B-104) and benzyl benzoate (Sigma B-6630). Whole embryos were mounted with the DV axis parallel to the cover slip (either animal pole up or down). Imaging was performed using a Ziess LSM880 confocal microscope with an LD LCI Plan-Achromat 25X/0.8 lmm Corr DIC M27 multi-immersion lens in the oil-immersion setting. A single bead from a calibration slide (ThermoFisher Scientific Cat#F369009, Well A1) was imaged between each slide of embryos to account for fluctuations in power of the 633 nm laser over time (This laser was used for detecting pSmad1/5, pSmad2, or HA/Flag). Immunoflourescence was performed as described in *Zinski et al., 2017* for pSmad2 immunostained embryos in *Figure 2* and pSmad1/5 Immunostained embryos in *Figure 4*. pSmad1/5 immunostained embryos in *Figures 2* and *5* were imaged with the following changes: embryos were imaged in a single ~567×567 µm frame and pixel dwell time was reduced to 0.77µsec. Receptor HA and Flag tagged embryos were imaged at 63x. Images of receptor stains were taken approximately 18 µm from the outer yolk syncytial layer of the embryo in a ~ 225×225 µm frame (*Figure 1a*). Quantitative receptor tag images were taken by imaging 16 2.2 µm stacks starting at the apical end of the embryo in a ~ 225×225 µm frame (*Figure 1—figure supplement 2*).

Embryo immunofluorescent intensity for pSmad1/5, pSmad2 or HA/Flag was normalized to a calibration bead imaged at the same time. Mean fluorescence of each fluorescent bead stacked image was measured using Imaris statistical analysis. Fold change in mean fluorescence was then calculated for each bead compared to a reference bead that was imaged immediately after an uninjected embryo was imaged. The fold change in mean fluorescence was then applied to the total fluorescence of each experimental-imaged embryo using the ImageJ multiply function.

For pSmad1/5 and pSmad2 immunostained embryos, calibrated Images were analyzed using the Imaris software spots function which identifies each nucleus within the immunostained embryo using the Sytox green nuclear stain and refined based on quality. Spot quality threshold is set using a control image to allow the program to detect all nuclei without detecting fluorescent points that are not nuclei, and then refined by eye (*Figure 2—source data 2*, *Figure 4n—source data 2*, *Figure 4t—source data 2*, *Figure 5—source data 3*). The Imaris program was then used to color each spot based on mean relative pSmad1/5 or pSmad2 (Alexa 647) intensity (AU) within the spot.

For HA immunostained embryos, calibrated embryos were analyzed using the Imaris software surfaces function to detect the area stained by β-catenin. The total HA flourescence was then measured within the area of β-catenin and the fluorescence per $\mu m^2$ was calculated (*Figure 1—figure supplement 2—source data 2*).

## Statistical analysis

For pSmad1/5 and pSmad2 immunostained embryos, mean fluorescence was calculated for all individual nuclei within each embryo using the Imaris spots function, as described above. For each embryo, nuclei that fell outside 1.8 times the interquartile range were removed. The mean of the remaining nuclei was calculated to generate a mean nuclear pSmad1/5 or pSmad2 fluorescence for each embryo. Mean nuclear fluorescence was then compared using a two-tailed T-test assuming unequal variance (*Figure 2—source data 1*, *Figure 4n—source data 2*, *Figure 4t—source data 2*, *Figure 5—source data 2*).

For receptor tag immunostained embryos, HA fluorescence per $\mu m^2$ calculated values that fell outside 1.8 times the interquartile range were excluded. HA fluorescence per $\mu m^2$ was also compared by a two-tailed T-test assuming unequal variance (*Figure 1—figure supplement 2—source data 2*).

Representative embryos in all figures were selected from the three embryos closest to the mean fluorescence of the group, using embryos without any significant tears or defects.

## In situ hybridization

Whole mount in situ hybridizations were performed on fixed 5–9 somite stage embryos using DIG-labeled anti-sense RNA probes (made with labeling kit: Roche 11277073910) to *pax2.1*, *krox20*, and *myod*. Probes were visualized with anti-DIG-Alkaline Phosphatase (Roche11093274910) and developed in BM Purple (Roche 11442074001). Embryos were mounted on agarose in methanol and photographed with a Leica IC80HD. Images were processed using image J. After imaging, embryos were collected in MeOH for genotyping. *pax2.1* and *krox20* expression patterns were categorized as completely radialized (as in a C5 embryo), expanded compared to wild-type, similar to wild-type, or restricted compared to wild-type. A representative embryo was chosen from the category most represented in each condition.

## Acknowledgements

We thank: the UPenn CDB Microscopy Core, especially Andrea Stout; Petra Seemann and Julia Haupt at Berlin-Brandenburg Center for Regenerative Therapies for providing the ligand-binding mutant receptor constructs; the UPenn fish facility; high school student Claire Tse; the Shore lab especially Meiqi Xu for ACVR1 constructs; Thomas Wardrop; Amy Kugath and members of the Mullins lab for helpful consultation. This work was supported by NIH R01-GM056326 and R35-GM131908 (to MCM) and R01-AR071399 (to EMS), a Developmental Grant from the Cali family and the Center for Research in FOP and Related Disorders, the Cali/Weldon Professorship (to EMS), and the International Fibrodysplasia Ossificans Progressiva Association (IFOPA).

## Additional information

### Funding

| Funder | Grant reference number | Author |
| --- | --- | --- |
| National Institute of General Medical Sciences | R35-GM131908 | Mary C Mullins |
| National Institute of General Medical Sciences | R01-GM056326 | Mary C Mullins |
| National Institute of Arthritis and Musculoskeletal and Skin Diseases | R01-AR071399 | Eileen M Shore |
| Developmental Grant from the Cali family | | Mary C Mullins |

| Center for Research in FOP and Related Disorders | Eileen M Shore |
|---|---|
| Cali/Weldon Professorship | Eileen M Shore |
| International Fibrodysplasia Ossificans Progressiva Association | Eileen M Shore |

The funders had no role in study design, data collection and interpretation, or the decision to submit the work for publication.

## Author contributions
Robyn S Allen, Conceptualization, Data curation, Formal analysis, Funding acquisition, Validation, Investigation, Visualization, Methodology, Writing - original draft, Writing - review and editing; Benjamin Tajer, Conceptualization, Validation, Investigation, Methodology, Writing - original draft, Writing - review and editing; Eileen M Shore, Mary C Mullins, Conceptualization, Data curation, Formal analysis, Supervision, Funding acquisition, Validation, Investigation, Visualization, Methodology, Writing - original draft, Project administration, Writing - review and editing

## Author ORCIDs
Robyn S Allen https://orcid.org/0000-0001-8306-0620
Benjamin Tajer https://orcid.org/0000-0001-7791-1249
Eileen M Shore https://orcid.org/0000-0003-2609-6971
Mary C Mullins https://orcid.org/0000-0002-9979-1564

## Ethics
Animal experimentation: This study was performed in strict accordance with the recommendations in the Guide for the Care and Use of Laboratory Animals of the National Institutes of Health. All of the animals were handled according to approved institutional animal care and use committee (IACUC) protocols (#803105) of the University of Pennsylvania.

## Decision letter and Author response
Decision letter https://doi.org/10.7554/eLife.53761.sa1
Author response https://doi.org/10.7554/eLife.53761.sa2

## Additional files

### Supplementary files
• Transparent reporting form

### Data availability
All data generated or analysed during this study are included in the manuscript and supporting files. Source data files have been provided for Figures 1, 3, 4, 5, 6, Figure 1-figure supplement 2, and Figure 5-figure supplement 2.

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
