## [Decision Letter]

**Acceptance summary:**

Fibrodysplasia Ossificans Progressiva (FOP) is a rare, but devastating genetic disease wherein connective tissue is turned to bone, progressively restricting movement by the patient. Mutations in the ACVR1 gene are causative for FOP, but many questions are unanswered about how connective tissue is turned to bone in this disease. In this study, a zebrafish embryonic patterning assay was used demonstrate that the FOP causing mutations in ACVR1 result in a receptor that can both respond to the normal ACVR1 ligand, non-normal ligands and can also act independent of ligand to mediate signaling. The authors also show that normal obligatory ACVR1 binding partners are not needed for signaling by this ACVR1 mutant receptor. This work represents a significant step forward in our understanding of the initiating steps that proceed the formation of bone outside of the skeleton in FOP.

**Decision letter after peer review:**

Thank you for submitting your article "FOP-ACVR1 Signals by Multiple Modalities in the Developing Zebrafish" for consideration by *eLife*. Your article has been reviewed by three peer reviewers, and the evaluation has been overseen by a Reviewing Editors and Kathryn Cheah as the Senior Editor. The following individuals involved in review of your submission have agreed to reveal their identity: Petra Knaus (Reviewer #3).

The reviewers have discussed the reviews with one another and the Reviewing Editor has drafted this decision to help you prepare a revised submission.

Summary:

Fibrodysplasia ossificans progressiva (FOP) is a very rare disease caused by mutations in the ACVR1 gene. This gene codes for a type I BMP/TGFβ receptor which signals via pSMAD1/5. There are two separate mutations that lead to FOP, R206H and G328R, both of which are activating mutations. This study, a zebrafish DV patterning assay is used to interrogate the mechanism by which ACVR1 mutants R206H and G328R promote canonical SMAD1/5-mediated signaling responses. Both FOP mutants with deletions in ligand binding domain were found to overactivate BMP/SMAD1/5-induced DV patterning. Moreover, whereas WT ACVR1 critically depends on BMPR1 for DV patterning, the data presented suggest that BMPR1 is dispensable for both BMP7-dependent and independent signaling by the ACVR R206H mutant. This latter finding is highly novel and considered the largest strength of this study.

Overall, this is well presented and coherently written paper. The introduction does a fantastic job of setting up the study. The experiment follows a logical order and the prior literature is considered both in the experimental design and interpretation of the results. The conclusions of this study are interesting and provocative, but in several instances, it was felt that these were overstated. Some additional experiments were requested by the reviewers and it was unanimously considered that these were essential for this manuscript to be accepted.

Essential revisions:

1) The concept of decreased/dispensability of ligand binding domain ACVR1 with FOP or constitutively active mutation were reported before by Haubt et al., 2014 using in vitro cultured cells. Allen et al., now validate and extend these previous results using in vivo zebrafish assays (as measured by DV patterning, pSMAD1/5 IF imaging and dorsal marker ISH staining). To extend the previous findings in vivo is an important step forward, but becomes even more stronger when complemented with in vitro culture/transfection experiments that are more easier to control and quantitate. The most novel finding by Allen et al., that the ΔACVR1-R206H can signal in the absence of all other type I receptors needs to be shown in in vitro cultured cells using BRE-luc and pSMAD1/5 assays and include (cell surface) expression controls (or lack thereof) for receptors. This experiment was considered essential by all reviewers.

2) The authors use mouse ALK2 for their studies in zebrafish. Mouse ALK2 differs quite a lot from Zebrafish ALK2 (the latter was first described in the literature as a novel type I receptor, ALK8). Would the authors have gotten the same results if Zebrafish ALK2 constructs would have been used; do mouse and zebrafish ALK2 form equally efficient complexes with endogenous zebrafish BMP type I and type II receptors? Or is it actually an advantage to use mouse ALK2 as it allows for better insights into role of mutated ALK2 in FOP, and uncouple it from effects from endogenous zebrafish activin ligands? It is unclear if mouse BMP7 or zebrafish BMP7 expression construct was used in their study.

3) FOP ALK2 mutants can mediate Activin-induced SMAD1/5 response. From that perspective it would have been interesting to examine the effect of Actin ectopic (instead of BMP7) expression. Is the response of ΔAcvr1-R206H also not responsive to activin?

4) All data was performed in zebrafish embryos in which WT and mutant receptors are likely overexpressed. Authors do not provide information whether the expression levels of ectopic receptor are comparable to that of endogenous receptors. Can the authors exclude the possibility that the observed results are not an artifact of overexpression?

5) The authors state that ACVR1-R206H is able to signal in the absence of all other BMP type I receptors. This is an overstatement, and needs to be corrected. Morpholinos are used, which will never completely deplete BMP receptor expression. In the experiments residual maternal BMP receptor expression is still present. ALK1 is also a BMP type I receptor that was not targeted for depletion by the authors.

6) Figure S3. Primary (imaging) data that ACVR-R206H is responsive to BMP7 ligand needs to be included. Include statistical analysis of the IF imaging data shown in Figure 2 and Figure 4.

7) The bubble graphs are an efficient manner of presenting the data. That said, can a legend or scaling system be devised such that we know what percent each sized bubble represents? This would aid with data transparency.

8) Figure 1I, column 5… this is only an incomplete rescue, not a full rescue as is stated in the text. This should be clarified.

9) It is difficult to see that there is indeed a gradient in Figure 2D,D’,E,E’. Can this be quantified in any way, both to see how reproducible this was and to make the read-out less subjective. A second concern is that the images for the FOP mutants are over saturated obscuring the pattern.

10) Using DV patterning assays in zebrafish embryos the group has convincingly shown that both Acvr1-R206H and Acvr1-G328R, two FOP mutants, are able to rescue an acvr1l deletion mutant, even when the ligand binding site is deleted. This is not seen in the corresponding wt-deletion mutant. In animal pole view experiment, in which pSMAD1/5 was measured, the ligand independent effects were confirmed. Here it would be of interest to also show data on pSMAD2/3 as it was suggested before that FOP-mutant receptors induce both branches of SMAD signaling. The authors should address this with the same assay shown in Figure 2. This experiment was considered essential by all reviewers.

11) Please explain the differences for R206H as compared to G328R in Figure 2: Δ-R206H acts stronger in the KD (e, e'), while Δ-G328R works stronger in the wt setting (g, g'). Is this statistically significant? What is the explanation for these differences?

12) The authors write in subsection “ACVR1-R206H signals in the absence of all other Type I BMP receptors”: "Therefore, the ability of FOP-ACVR1 to signal independently of ligand suggests the possibility of signaling in the absence of receptor complex partners as well." This statement is interesting; however, the absence of receptor complex partners is not the only explanation for this observation. In particular, the authors should refer to and/or address the role of the low affinity receptor for BMP (BMPR2) or the high affinity receptor for Activin (ActR2) in this context. The data on the contribution of BMPR1a and BMPR1b however are convincingly shown and important as they give evidence that for the ligand-independent signaling by the FOP-mutant receptor, none of these type 1 receptors are required. It would be however important to mention the expression of other type 1 receptors such as ALK4 and ALK5 as potentially compensating receptors. This is of particular importance also for point 1 (SMAD2/3 signaling), mentioned above.

13) The authors continued to show, that Acvr1-R206H responds to BMP7 in the absence of BMPR1A and BMPR1B. It would be of particular interest to extend this also to other ligands. While testing all ligands is beyond the scope of this paper, this topic should be addressed in the discussion as potential needed future work and or in context with that which is already know.

14) In the Discussion section the authors write: "….further suggests it does not have the ability to bind ligand, and therefore does not need ligand to signal." This is a strong statement, in particular, as others have shown that Activin binding to the mutant receptors leads to increased SMAD signaling. How do the authors explain these controversial findings?

---

## [Author Response]

Summary:Fibrodysplasia ossificans progressiva (FOP) is a very rare disease caused by mutations in the ACVR1 gene. This gene codes for a type I BMP/TGFβ receptor which signals via pSMAD1/5. There are two separate mutations that lead to FOP, R206H and G328R, both of which are activating mutations. This study, a zebrafish DV patterning assay is used to interrogate the mechanism by which ACVR1 mutants R206H and G328R promote canonical SMAD1/5-mediated signaling responses. Both FOP mutants with deletions in ligand binding domain were found to overactivate BMP/SMAD1/5-induced DV patterning. Moreover, whereas WT ACVR1 critically depends on BMPR1 for DV patterning, the data presented suggest that BMPR1 is dispensable for both BMP7-dependent and independent signaling by the ACVR R206H mutant. This latter finding is highly novel and considered the largest strength of this study.Overall, this is well presented and coherently written paper. The introduction does a fantastic job of setting up the study. The experiment follows a logical order and the prior literature is considered both in the experimental design and interpretation of the results. The conclusions of this study are interesting and provocative, but in several instances, it was felt that these were overstated. Some additional experiments were requested by the reviewers and it was unanimously considered that these were essential for this manuscript to be accepted.Essential revisions:1) The concept of decreased/dispensability of ligand binding domain ACVR1 with FOP or constitutively active mutation were reported before by Haubt et al., 2014 using in vitro cultured cells. Allen et al., now validate and extend these previous results using in vivo zebrafish assays (as measured by DV patterning, pSMAD1/5 IF imaging and dorsal marker ISH staining). To extend the previous findings in vivo is an important step forward, but becomes even more stronger when complemented with in vitro culture/transfection experiments that are more easier to control and quantitate. The most novel finding by Allen et al. that the ΔACVR1-R206H can signal in the absence of all other type I receptors needs to be shown in in vitro cultured cells using BRE-luc and pSMAD1/5 assays and include (cell surface) expression controls (or lack thereof) for receptors. This experiment was considered essential by all reviewers.

As noted by the reviewers, ligand-independent signaling through FOP ACVR1 has been reported previously, however some in the field have not found the reported in vitro data to be fully convincing that this is an established mechanism of action for the mutant ACVR1 receptor. Therefore, our in vivo studies are an important advance in understanding FOP and the mechanisms of BMP receptor regulation and moving the field forward.

We agree with the reviewers that the suggested additional in vitro experiments would provide additional in vitro evidence to support our in vivo data, and in response to the reviewers we designed experiments to conduct these studies. We had planned to initiate these experiments in April, however, our labs were closed in mid-March in response to the COVID-19 pandemic. We gained limited access to our labs in mid-June, just this week transitioning from 20% occupancy. At this time, our cell culture facilities are not fully operational and we still do not have the ability to conduct these experiments. Once we are able to access the facility, we expect these experiments would take a minimum of three months to complete. In addition, the first author of this paper has re-entered her clinical year of veterinary school as part of her dual VMD-PhD program, and has limited access to the lab at this time. And while we agree that performing these experiments in in vitro cell culture would extend these studies, we believe the added controls and quantitation of our in vivo data in this revision are very strong and compelling independently of repeating the work in cell culture.

In particular, we would like to address the concern that our in vivo data are not as well controlled or quantifiable as cell culture. With the reviewer’s suggestions, we have added quantification of pSmad1/5 signaling in our immunostained embryos and performed statistical analysis on these data (Figure 2H, Figure 4N,T and Figure 5K). We hope they demonstrate the power of our system in detecting changes in signaling intensity by BMP receptors. In addition, we have further clarified in the body of the text our use of reproducible pSmad1/5 signaling-dose dependent phenotypes to verify knockdown of different receptors in our system (Results section and Discussion section). *bmp7* mRNA overexpression in an embryo normally over-activates pSmad1/5 signaling resulting in a ventralized phenotype (V1-5). Acvr1l KD produces the most dorsalized phenotype, C5, and cannot be rescued by *bmp7* mRNA overexpression, demonstrating that this receptor is necessary for signaling, and knockdown was sufficient to prevent response to Bmp7 ligand. In one of our experiments, the knockdown was not complete in all embryos; we have repeated this experiment to obtain complete knockdown (Figure 1E,F,G). These data are further supported by our pSmad1/5 immunostaining results showing Acvr1l KD results in loss of pSmad1/5 signaling that cannot be rescued by *bmp7* overexpression. Bmpr1b KD has no effect on wildtype fish, but consistently dorsalizes *bmpr1a*-/- fish to a C5 phenotype even in the presence of endogenous Bmp7, indicating that knockdown eliminated pSmad1/5 signaling. We recognize that we previously did not thoroughly explain these and other controls in the manuscript, and this is now added.

2) The authors use mouse ALK2 for their studies in zebrafish. Mouse ALK2 differs quite a lot from Zebrafish ALK2 (the latter was first described in the literature as a novel type I receptor, ALK8). Would the authors have gotten the same results if Zebrafish ALK2 constructs would have been used; do mouse and zebrafish ALK2 form equally efficient complexes with endogenous zebrafish BMP type I and type II receptors? Or is it actually an advantage to use mouse ALK2 as it allows for better insights into role of mutated ALK2 in FOP, and uncouple it from effects from endogenous zebrafish activin ligands? It is unclear if mouse BMP7 or zebrafish BMP7 expression construct was used in their study.

This study uses both mouse and human ACVR1 (ALK2). Zebrafish Acvr1l (ALK8) has 69% identity with the amino acid sequence of human ACVR1 overall and 85% identity in the intracellular domain. We chose to use mouse and human ACVR1 because of their greater relevance to the human disease, Fibrodysplasia ossificans progressiva. Both mouse and human WT-ACVR1 are able to rescue loss of endogenous zebrafish Acvr1l, showing that they can replace its function, including its role in the BMP receptor complex. This is further supported in our studies by the ability of both mouse and human WT-ACVR1 to respond to zebrafish Bmp7 (Figure 3).

3) FOP ALK2 mutants can mediate Activin-induced SMAD1/5 response. From that perspective it would have been interesting to examine the effect of Actin ectopic (instead of BMP7) expression. Is the response of ΔAcvr1-R206H also not responsive to activin?

In response to the reviewers’ question we have now tested the ability of both ACVR1-R206H and ΔACVR1-R206H to respond to Activin A using our pSmad1/5 quantitative immunofluorescence protocol and included these data in our manuscript (new Figure 4). We show that ACVR1-R206H, but not ΔACVR1-R206H, is able to respond to both Bmp7 and to Activin A.

Note that because Activin A also activates the Nodal pathway in developing zebrafish embryos, phenotypic evaluation of Activin A response in later stage embryos is not possible. Embryos arrest at epiboly and die before DV patterning or, in rare cases, are phenotypically uninterpretable and cannot be assessed (Figure 4A,B).

4) All data was performed in zebrafish embryos in which WT and mutant receptors are likely overexpressed. Authors do not provide information whether the expression levels of ectopic receptor are comparable to that of endogenous receptors. Can the authors exclude the possibility that the observed results are not an artifact of overexpression?

We agree that this is an important point that we had not addressed in our experiments. We now compared the level of mouse WT-ACVR1 that was sufficient to just rescue Acvr1ldepleted embryos to the amount of ACVR1-R206H, ΔACVR1-R206H, ACVR1-G328R, and ΔACVR1-G328R that ventralized Acvr1l-depleted embryos (Figure 1—figure supplement 2). We used quantitative immunofluorescence of antibody binding to the HA-epitope tag on these receptors, quantitating immunostaining at the cell membrane via co-localization with betaCatenin immunofluorescence. Our results show that similar or less amounts of ACVR1-R206H, ΔACVR1-R206H, ACVR1-G328R, and ΔACVR1-G328R protein are required to ventralize the embryo compared to the amount of WT-ACVR1 protein that is required to rescue loss of endogenous Acvr1l. Thus, our results cannot be explained by an overexpression artifact. These new results have been added to the manuscript (Results section).

We also note that the amount of ACVR1-R206H mRNA injected to achieve a ventralized phenotype in this study was 1/20^th^ the dose of WT-ACVR1 mRNA required to rescue the loss of endogenous WT-Acvr1l. While the amount of mRNA does not necessarily correlate with the amount of protein produced, the large degree to which these concentrations differ suggests that significantly less ACVR1-R206H is required to ventralize the zebrafish embryo than the amount of WT-ACVR1 required to rescue the embryo, and as a result is expected to be less than the endogenous amount of Acvr1l protein.

5) The authors state that ACVR1-R206H is able to signal in the absence of all other BMP type I receptors. This is an overstatement, and needs to be corrected. Morpholinos are used, which will never completely deplete BMP receptor expression. In the experiments residual maternal BMP receptor expression is still present. ALK1 is also a BMP type I receptor that was not targeted for depletion by the authors.

We agree that this was an overstatement and have now revised our manuscript to state that ACVR1-R206H is able to signal independently of Bmpr1 specifically.

We believe our results strongly indicate that Bmpr1, a required partner for WT-ACVR1, is not required for ACVR1-FOP-mediated patterning. Our Bmpr1, Acvr1l triple

knockdown/mutant embryo is unable to respond to Bmp7 overexpression, showing that there is not enough receptor present following morpholino knockdown to be relevant to signal in patterning. We note that females are *bmpr1aa*+/-, so are depleted of maternal *bmpr1aa* as well. Further, *bmpr1a*-/- mutant embryos are unable to respond to Bmp7 overexpression (Figure 7A columns 3,10,11), indicating that the half-dose of maternal *bmpr1aa* is insufficient to signal in DV patterning. It is possible that this residual transcript is degraded shortly after the initiation of zygotic transcription at 3 hpf, when a large fraction of maternal transcripts are degraded, a timepoint 2.5 hours before BMP signaling patterns the zebrafish embryo.

It is also important to note that AcvrlI (ALK1) is not present in the embryo at the time of our pSmad1/5 immunostains. Studies have shown by RNA seq that *acvrlI* transcript is not detectable until 75% epiboly^1^ and is not detectable by in situ during the first 24 hours of development^2^. We cannot exclude, however, that low levels of AcvrlI expression play a role in the ability of ACVR1-R206H to respond to ligand in the absence of Bmpr1 and Acvr1, as shown in Figure 7. We thank the reviewers for raising these questions and these points have now been added to the Discussion section.

6) Figure S3. Primary (imaging) data that ACVR-R206H is responsive to BMP7 ligand needs to be included. Include statistical analysis of the IF imaging data shown in Figure 2 and Figure 4.

Figure S3 has been converted to the second part of Figure 3 within the main text and images of embryo phenotypes have been added as supplemental figures (Figure 3—figure supplement 2).

For all pSmad1/5 immunofluorescence imaging (Figure 2, Figure 4, and Figure 5), we have now added plots comparing the means of the mean nuclear fluorescence of each embryo in an experimental condition. Two tailed T-tests assuming unequal variance have been performed on these data and the results have been added. Note that the number of embryos analyzed in Figure 2 and Figure 5 has changed since our last submission. More embryos were immunostained and a few had to be removed from analysis due to a non-specific stain in the outer cell layer.

7) The bubble graphs are an efficient manner of presenting the data. That said, can a legend or scaling system be devised such that we know what percent each sized bubble represents? This would aid with data transparency.

Thank you for this suggestion. We have added a legend in Figure 1 for the bubble plots showing the size of example bubbles that correspond to 100, 90, 80, 70, 60, 50, 40, 30, 20 and 10% of the total population.

8) Figure 1I, column 5… this is only an incomplete rescue, not a full rescue as is stated in the text. This should be clarified.

We have restated our results to state that Acvr1 primarily rescues *acvr1l* KD embryos to mildly dorsalized or wildtype phenotypes (Results section).

9) It is difficult to see that there is indeed a gradient in Figure 2D,D’,E,E’. Can this be quantified in any way, both to see how reproducible this was and to make the read-out less subjective. A second concern is that the images for the FOP mutants are over saturated obscuring the pattern.

We agree with the reviewers’ point. The results from our immunofluorescent studies have now been composited in graphs showing the means of the mean nuclear fluorescence of each embryo in an experimental condition (Figure 2, Figure 4 and Figure 5), which greatly strengthens this analysis. In many of these embryos, the visible gradient is lost because signaling is high in all nuclei, as is the case for d, d’, e, and e’. All embryos being compared to each other have been normalized and processed in the same way, allowing us to compare the loss of gradient and bright signal to the normal gradient in wildtype un-injected embryos.

10) Using DV patterning assays in zebrafish embryos the group has convincingly shown that both Acvr1-R206H and Acvr1-G328R, two FOP mutants, are able to rescue an acvr1l deletion mutant, even when the ligand binding site is deleted. This is not seen in the corresponding wt-deletion mutant. In animal pole view experiment, in which pSMAD1/5 was measured, the ligand independent effects were confirmed. Here it would be of interest to also show data on pSMAD2/3 as it was suggested before that FOP-mutant receptors induce both branches of SMAD signaling. The authors should address this with the same assay shown in Figure 2. This experiment was considered essential by all reviewers.

We have now performed pSmad2 immunostaining on *bmp7*-/- embryos not injected

(which have normal Nodal signaling and as a result normal pSmad2 expression) or injected with *Activin A*, *Acvr1-R206H* or *Acvr1-R206H* plus *Activin A* mRNA. While Activin A increases pSmad2 expression, presumably by activating the Nodal signaling pathway, Acvr1-R206H does not increase pSmad2 expression. Acvr1-R206H plus Activin A does increase pSmad2 signaling, but not any more than Activin A alone. These data are now included in Figure 4O-T.

11) Please explain the differences for R206H as compared to G328R in Figure 2: Δ-R206H acts stronger in the KD (e, e'), while Δ-G328R works stronger in the wt setting (g, g'). Is this statistically significant? What is the explanation for these differences?

Both ACVR1-R206H and ACVR1-G328R produce a range of DV phenotypes (as indicated in bubble plots), and this is reflected in a range of pSmad1/5 intensities. Quantification of pSmad1/5 intensity has now been added to provide more information (Figure 2H). The embryos shown in Figure 2A-G, Figure 4C-M, 4O-S, and Figure 5G-J were selected from the 3 embryos closest to the mean fluorescence of the condition and did not have any significant defects or tears. These selection criteria have been added to the Materials and methods section.

In Figure 2, with the exception of ΔAcvr1, none of the receptors induce a significant difference in pSmad1/5 intensity with verses without Acvr1l KD (Figure 2H). In general, the mean intensity of pSmad1/5 appears to decrease with *acvr1l* KD. This is most likely due to the lower baseline signaling activity in the *acvr1l* KD embryos before the activity of FOP-ACVR1.

12) The authors write in subsection “ACVR1-R206H signals in the absence of all other Type I BMP receptors”: "Therefore, the ability of FOP-ACVR1 to signal independently of ligand suggests the possibility of signaling in the absence of receptor complex partners as well." This statement is interesting; however, the absence of receptor complex partners is not the only explanation for this observation. In particular, the authors should refer to and/or address the role of the low affinity receptor for BMP (BMPR2) or the high affinity receptor for Activin (ActR2) in this context. The data on the contribution of BMPR1a and BMPR1b however are convincingly shown and important as they give evidence that for the ligand-independent signaling by the FOP-mutant receptor, none of these type 1 receptors are required. It would be however important to mention the expression of other type 1 receptors such as ALK4 and ALK5 as potentially compensating receptors. This is of particular importance also for point 1 (SMAD2/3 signaling), mentioned above.

We thank the reviewers for noting that our “data on the contribution of BMPR1a and BMPR1b however are convincingly shown and important as they give evidence that for the ligand-independent signaling by the FOP-mutant receptor, none of these type 1 receptors are required.” We agree with the reviewers that we have not excluded relevant roles of type II receptors or Alk4 and 5 receptors. We have modified the Discussion section to more thoroughly address the role that the type II receptors may play. Previously, we discussed the possibility that ACVR1-R206H may be able to signal with only type II BMP receptors. We have now also addressed that the type II receptors that signal in this case may depend on the ligand available.

Previously, we also discussed the possibility that ACVR1 may be signaling in combination with TGFB1 (Alk5) and ACVR1b (Alk4). We have further added to the Discussion section that the preference for these receptors may be influenced by the associated ligand and type II receptor, in particular in the case of ACVR1b.

13) The authors continued to show, that Acvr1-R206H responds to BMP7 in the absence of BMPR1A and BMPR1B. It would be of particular interest to extend this also to other ligands. While testing all ligands is beyond the scope of this paper, this topic should be addressed in the discussion as potential needed future work and or in context with that which is already know.

As suggested, this has been added to the Discussion section as a future direction for this work.

14) In the Discussion section the authors write: "….further suggests it does not have the ability to bind ligand, and therefore does not need ligand to signal." This is a strong statement, in particular, as others have shown that Activin binding to the mutant receptors leads to increased SMAD signaling. How do the authors explain these controversial findings?

This statement previously referred specifically to ΔAcvr1-R206H, which lacks a large part of the ligand binding domain. Unlike Acvr1-R206H, ΔAcvr1-R206H does not signal in response to either Bmp7 or Activin A. This has been expanded and further clarified in the new Figure 4 and Results section . The results suggest that Acvr1-R206H can signal without binding and responding to ligand.